# Filling gaps in bacterial catabolic pathways with computation and high-throughput genetics

**Morgan N. Price** *, **Adam M. Deutschbauer** , **Adam P. Arkin** *

Environmental Genomics and Systems Biology, Lawrence Berkeley National Laboratory, Berkeley, California, United States of America

* funwithwords26@gmail.com (MNP); aparkin@lbl.gov (APA)

## Abstract

To discover novel catabolic enzymes and transporters, we combined high-throughput genetic data from 29 bacteria with an automated tool to find gaps in their catabolic pathways. GapMind for carbon sources automatically annotates the uptake and catabolism of 62 compounds in bacterial and archaeal genomes. For the compounds that are utilized by the 29 bacteria, we systematically examined the gaps in GapMind's predicted pathways, and we used the mutant fitness data to find additional genes that were involved in their utilization. We identified novel pathways or enzymes for the utilization of glucosamine, citrulline, myo-inositol, lactose, and phenylacetate, and we annotated 299 diverged enzymes and transporters. We also curated 125 proteins from published reports. For the 29 bacteria with genetic data, GapMind finds high-confidence paths for 85% of utilized carbon sources. In diverse bacteria and archaea, 38% of utilized carbon sources have high-confidence paths, which was improved from 27% by incorporating the fitness-based annotations and our curation. GapMind for carbon sources is available as a web server (http://papers.genomics.lbl.gov/carbon) and takes just 30 seconds for the typical genome.

**Data Availability Statement:** The code for GapMind, including the rules that describe carbon catabolism, is available in the PaperBLAST code base (https://github.com/morgannprice/PaperBLAST). The code and the analysis results

## Author summary

For many microbes, we know little about them beyond their genome sequences. In principle, we could use genome sequences to predict microbes' traits, such as which carbon sources they can eat, but first we need to identify more of the genes involved. We built an automated tool, GapMind, to annotate the transporters and enzymes for utilizing 62 common carbon sources, and used GapMind to identify gaps: transporters or enzymes that should be present, to explain how a bacterium uses a carbon source, but could not be found in the genome. By comparing these gaps to large-scale genetic data for 29 bacteria, we identified hundreds of novel transporters and enzymes, and a new metabolic pathway for consuming glucosamine. When we added these novel genes to GapMind, its results for diverse bacteria and archaea improved significantly.

are also archived (https://doi.org/10.6084/m9.figshare.16906993.v1). All of the fitness data we analyzed is available in the fitness browser (http://fit.genomics.lbl.gov/) or for download (https://doi.org/10.6084/m9.figshare.16913530.v1).

**Funding:** This material by ENIGMA- Ecosystems and Networks Integrated with Genes and Molecular Assemblies (http://enigma.lbl.gov), a Science Focus Area Program at Lawrence Berkeley National Laboratory is based upon work supported by the U.S. Department of Energy, Office of Science, Office of Biological & Environmental Research under contract number DE-AC02-05CH11231. The funders had no role in study design, data collection and analysis, decision to publish, or preparation of the manuscript.

**Competing interests:** The authors have declared that no competing interests exist.

## Introduction

Genome sequences are now available for tens of thousands of bacterial species [1], and for most of these bacteria, little else is known about them. In principle, the genome sequence could allow us to predict the capabilities of the organism, such as what nutrients it can use, but in practice this is challenging. For instance, metabolic models can be generated automatically from a genome sequence, and these metabolic models can be used to predict which carbon sources the organism can grow on, but these predictions are only 50–70% accurate [2,3]. More accurate predictions are not currently feasible because annotations of the functions of transporters and enzymes are often erroneous [4,5] and because new families of transporters and enzymes and new catabolic pathways continue to be discovered. Also, even if the genome contains genes for the necessary proteins, the proteins might not be expressed.

To discover novel catabolic enzymes and transporters on a large scale, we used a combination of large-scale mutant fitness data and computation. First, we built an automated tool to annotate catabolic pathways. GapMind for carbon sources uses a similar approach as GapMind for amino acids [6]. GapMind relies on known pathways (mostly from MetaCyc [7]) and a database of experimentally-characterized proteins. Given a genome and a carbon source, GapMind identifies the most plausible pathway for consuming the compound, and it highlights any gaps.

Next, we used large-scale mutant fitness data from 29 heterotrophic bacteria [4,8] to try and fill these gaps. For each of these bacteria, a pool of tens of thousands of barcoded transposon mutants was grown in various defined media and the change in each mutant's abundance was quantified by DNA sequencing. If the initial version of GapMind (developed without using the fitness data) had any gaps, we tried to fill the gaps by using genes that were important for fitness during growth on that carbon source, but were not important in most other conditions. Using this approach, we identified functions for hundreds of diverged proteins. Highlights include a new pathway for the utilization of glucosamine; a new family of citrullinases; a new family of aldolases that are involved in myo-inositol catabolism; the first identification of genes for 3'-ketolactose hydrolases, which are involved in lactose catabolism; and a novel oxepin-CoA hydrolase for phenylacetate catabolism. By using PaperBLAST to find papers about homologs of the candidate genes [9], we also identified over 100 relevant proteins that were experimentally characterized but whose function was not described in curated databases such as Swiss-Prot [10], BRENDA [11], MetaCyc [7], CAZy [12], or TCDB [13].

We incorporated all of these additional enzymes and transporters into GapMind, and we asked how much the coverage of catabolism in diverse bacteria and archaea had improved. We relied on the IJSEM database, which reports carbon sources utilized by diverse bacteria and archaea [14]. (The International Journal of Systematic and Environmental Microbiology publishes species descriptions, which often report carbon sources that are utilized by the type strain.) Across diverse bacteria and archaea with sequenced genomes, coverage by high-confidence paths was improved by 11% (from 27% to 38%) after the incorporation of annotated and curated proteins into GapMind. We also used the fitness data from the 29 heterotrophic bacteria to confirm that GapMind usually selects the correct pathway and genes for utilizing each carbon source. Overall, we filled many gaps in carbon catabolism, and we improved our understanding of catabolism in diverse prokaryotes significantly, but much remains to be discovered.

## Results and discussion

### Overview of GapMind for carbon sources

GapMind describes the utilization of 62 carbon sources, including 19 amino acids, 19 simple sugars or sugar acids, 5 disaccharides, and 11 organic acids (Fig 1). GapMind describes the

**Amino acids**
D-alanine
L-alanine
L-arginine
L-asparagine
L-aspartate
citrulline
L-glutamate
L-histidine
L-isoleucine
L-leucine
L-lysine
L-phenylalanine
L-proline
D-serine
L-serine
L-threonine
L-tryptophan
L-tyrosine
L-valine

**Organic acids**
acetate
citrate
fumarate
D-lactate
L-lactate
L-malate
2-oxoglutarate
phenylacetate
propionate
pyruvate
succinate

**Simple sugars**
L-arabinose
2-deoxyribose
D-fructose
L-fucose
D-galactose
D-galacturonate
D-gluconate
D-glucosamine
D-glucose
D-glucuronate
D-mannitol
D-mannose
myo-inositol
N-acetylglucosamine
L-rhamnose
D-ribose
D-sorbitol
xylitol
D-xylose

**Disaccharides**
cellobiose
lactose
maltose
sucrose
trehalose

**Other**
deoxyinosine
deoxyribonate
ethanol
D-glucose 6-phosphate
glycerol
4-hydroxybenzoate
putrescine
thymidine

**Fig 1. The 62 carbon sources described in GapMind.**

uptake of each compound and enzymatic transformation until the compound reaches central metabolism. For the catabolism of the standard amino acids, GapMind does not describe transamination reactions, such as the conversion of L-alanine to pyruvate, because these transaminases tend to be non-specific and genetically redundant (see [6]). For central intermediates such as pyruvate, GapMind only describes their uptake; similarly, for L-alanine and L-aspartate, which are converted to central metabolites by transamination, GapMind describes only their uptake. More broadly, GapMind does not represent central metabolism or the production of ATP; for instance, GapMind represents the utilization of acetate by its uptake and conversion to acetyl-CoA, but not the generation of energy from acetyl-CoA (such as by the tricarboxylic acid cycle and the glyoxylate shunt). GapMind only includes pathways that yield fixed carbon and hence allow growth, so many fermentative pathways that yield energy and by-products are not included. For instance, some anaerobic bacteria can ferment leucine to isovalerate (3-methylbutanoate), isocaproate (4-methylpentanoate), and $CO_2$; this process generates energy but does not yield any fixed carbon, and is not represented in GapMind. GapMind also does not represent uptake through outer membrane porins: porins are often non-specific (as in *Escherichia coli*) or unnecessary (as in most archaea and Firmicutes).

GapMind describes the utilization of carbon sources with 1,309 steps, where each step corresponds to a group of proteins that have the same function as an enzyme, a transporter, or a component thereof. (Enzymes and transporters with multiple subunits are represented with one "step" per subunit.) 493 steps are enzymes and 816 steps are transporters. These steps are represented by the sequences of 6,742 experimentally-characterized proteins and by 164 hidden Markov models of protein families from TIGRFAMs [15]. Most of these functionally-

characterized proteins are from curated databases, but 11% were identified from fitness data while building GapMind, and 2% were curated from the literature while building GapMind (Table 1). 56% of these proteins are from bacteria, 6% are from archaea, and 38% are from eukaryotes.

Based on these steps, GapMind describes the utilization of each carbon source with alternate rules. For example, pyruvate can be transported by nine different types of transporters, three of which have more than one component. Most of the carbon sources can be degraded by more than one metabolic pathway: the exceptions are deoxyribonate, D-lactate, L-leucine, L-serine, L-tyrosine, and seven carbon sources for which only transport is represented.

Given the steps and potential pathways, and a genome of interest, GapMind searches for candidates for each step, and then selects the best path for the utilization of each carbon source (Fig 2). This aspect of GapMind for carbon sources is almost unchanged from GapMind for amino acid biosynthesis [6]. Briefly, GapMind searches the predicted proteins for candidates by using ublast (a fast alternative to protein BLAST) or HMMer [16,17]. A candidate from ublast is considered high-confidence if it is at least 40% identical (amino acid sequence) to a characterized protein, the alignment has at least 80% coverage, and the candidate is more similar to proteins known to perform this step than to characterized proteins with other functions. Similarity near this threshold (40–50% identity and 80% coverage) corresponds to an estimated 73% accuracy for annotating enzymes, but just 32% accuracy for annotating transporters (see Materials and Methods). Other candidates from ublast are medium-confidence if they are at least 30% identical with 80% coverage and are less similar to characterized proteins with other functions, or if they are at least 40% identical with 70% coverage (regardless of similarity to proteins with other functions). A candidate from HMMer is considered high-confidence if the alignment covers 80% of the model and the protein is not too similar to proteins with other functions (no alignment with 40% identity and 80% coverage). Given the confidence level for each step, GapMind looks for a path that has all high-confidence steps, or has no low-confidence steps, or has the highest total score. (Each high-confidence step scores +1, each medium-confidence step scores -0.1, and each low-confidence step scores -2.) GapMind for carbon sources typically takes about 30 seconds to analyze a genome.

We will first describe the novel biology we discovered while building GapMind, and then assess the quality of its results.

**Table 1. The sources of the experimentally-characterized proteins that perform the steps in GapMind.** The total is less than the sum of the entries because many proteins appear in more than one database.

| Source | Proteins |
|---|---|
| Swiss-Prot (characterized subset) | 2,421 |
| BRENDA | 2,099 |
| MetaCyc | 1,137 |
| CAZy | 1,108 |
| TCDB | 766 |
| From fitness data (this study) | 716 |
| CharProtDB | 435 |
| From fitness data (previous) | 428 |
| EcoCyc | 331 |
| From literature (this study) | 125 |
| *Total* | *6,742* |

## catabolism of small carbon sources in Pseudomonas fluorescens FW300-N2E2

Pathways are sorted by completeness. Sort by name instead.

| Pathway | Steps |
|---|---|
| valine | **livF**, **livG**, **livJ**, **livH**, **livM**, **bkdA**, **bkdB**, **bkdC**, **lpd**, **acdH**, **ech**, **bch**, **mmsB**, **mmsA**, **prpC**, **acnD**, **prpF**, **acn**, **prpB** |
| isoleucine | **livF**, **livG**, **livJ**, **livH**, **livM**, **bkdA**, **bkdB**, **bkdC**, **lpd**, **acdH**, **ech**, **ivdG**, **fadA**, **prpC**, **acnD**, **prpF**, **acn**, **prpB** |
| leucine | **livF**, **livG**, **livJ**, **livH**, **livM**, **ilvE**, **bkdA**, **bkdB**, **bkdC**, **lpd**, **liuA**, **liuB**, **liuD**, **liuC**, **liuE**, **atoA**, **atoD**, **atoB** |
| phenylalanine | **livF**, **livG**, **livH**, **livM**, **livJ**, **PAH**, **PCBD**, **QDPR**, **HPD**, **hmgA**, **maiA**, **fahA**, **atoA**, **atoD**, **atoB** |
| lysine | **argT**, **hisM**, **hisQ**, **hisP**, **davB**, **davA**, **davT**, **davD**, **gcdG**, **gcdH**, **ech**, **fadB**, **atoB** |
| threonine | **braC**, **braD**, **braE**, **braF**, **braG**, **ltaE**, **adh**, **ackA**, **pta**, **gcvP**, **gcvT**, **gcvH**, **lpd** |
| arginine | **artJ**, **artM**, **artP**, **artQ**, **arcA**, **arcB**, **arcC**, **aruF**, **aruG**, **astC**, **astD**, **astE** |
| citrulline | **AO353_03055**, **AO353_03050**, **AO353_03045**, **AO353_03040**, **arcB**, **arcC**, **aruF**, **aruG**, **astC**, **astD**, **astE** |
| myoinositol | **PS417_11885**, **PS417_11890**, **PS417_11895**, **iolG**, **iolE**, **iolD**, **iolB**, **iolC**, **iolJ**, **mmsA**, **tpi** |
| 4-hydroxybenzoate | **pcaK**, **pobA**, **pcaH**, **pcaG**, **pcaB**, **pcaC**, **pcaD**, **catI**, **catJ**, **pcaF** |

⋮

| deoxyribonate | deoxyribonate-transport, deoxyribonate-dehyd, ketodeoxyribonate-cleavage, **garK**, **atoA**, **atoD**, **atoB** |
| tryptophan | **aroP**, tnaA |
| deoxyinosine | nupC, **deoD**, deoB, deoC, **adh**, **ackA**, **pta** |
| thymidine | nupG, **deoA**, deoB, deoC, **adh**, **ackA**, **pta** |
| D-serine | cycA, dsdA |
| xylitol | PLT5, xdhA, **xylB** |
| rhamnose | rhaT, LRA1, LRA2, **LRA3**, **LRA4**, **aldA** |
| NAG | nagEcba, nagA, nagB |
| fucose | **HSERO_RS05250**, HSERO_RS05255, HSERO_RS05260, fucU, fucI, fucK, fucA, **tpi**, **aldA** |
| phenylacetate | ppa, paaK, paaA, paaB, paaC, paaE, **paaG**, paaZ1, paaZ2, **paaJ1**, **paaF**, **paaH**, **paaJ2** |

Confidence: **high confidence** medium confidence low confidence

transporter – transporters and PTS systems are shaded because predicting their specificity is particularly challenging.

**Fig 2. Example results for *Pseudomonas fluorescens* FW300-N2E2.** A page on the GapMind website shows the 62 compounds in order; this figure shows screenshots for the first 10 (highest-scoring) and last 10 (lowest-scoring) carbon sources. Hovering on a step shows the description and the best candidate, if any. (Some transporter components are named by the genes' locus tags; none of these locus tags are from *P. fluorescens* FW300-N2E2 itself.) Clicking on a step shows all the candidates for that step. Clicking on a compound shows alternate pathways.

### Glucosamine utilization via putative transmembrane transacetylase NagX

Fitness data from five diverse bacteria showed that the protein NagX is involved in the utilization of glucosamine as the sole source of carbon or nitrogen (Fig 3A–3E). The NagX family of transmembrane proteins is often found in operons for chitin utilization [18], but its function is not known. In four of the five bacteria, we found that N-acetylglucosamine 6-phosphate

### (A) *Shewanella sp.* ANA-3

| Shewana3 | | glucosamine (carbon) | | NAcGln (carbon) | | D,L-lactate (carbon) | |
|---|---|---|---|---|---|---|---|
| _3111 | *nagX* | -5.5 | -5.1 | 0.0 | -0.0 | 0.1 | 0.1 |
| _3110 | *nagP* | -4.8 | -6.5 | -3.3 | -3.4 | 0.1 | 0.2 |
| _3114 | *nagK* | -6.7 | -6.1 | -4.7 | -5.3 | -0.1 | -0.2 |
| _3112 | *nagA* | -6.2 | -5.7 | -6.0 | -6.5 | 0.2 | 0.1 |
| _3113 | *nagB* | -6.4 | -6.7 | -7.8 | -4.7 | 0.1 | 0.1 |

### (E) *Echinicola vietnamensis* KMM 6221

| Echvi | | glucosamine (carbon) | | | glucosamine (nitrogen) | | glucose (carbon) | |
|---|---|---|---|---|---|---|---|---|
| _1106 | *nagX* | -3.6 | -3.5 | -3.4 | -1.2 | -1.4 | -0.1 | 0.1 |
| _1226 | *transporter* | -4.5 | -4.1 | -4.3 | -2.0 | -1.7 | -0.1 | -0.1 |
| _3893 | *nagB?* | -4.6 | -4.7 | -4.0 | -4.2 | -3.9 | 0.1 | 0.2 |
| _1268 | *acs?* | -3.6 | -3.8 | -3.7 | -0.9 | -1.0 | 0.0 | -0.0 |

### (B) *Shewanella amazonensis* SB2B

| Sama | | glucosamine (nitrogen) | | NAcGln (carbon) | | | | NAcGln (nitrogen) | | D,L-lactate (carbon) | |
|---|---|---|---|---|---|---|---|---|---|---|---|
| _0947 | *nagX* | -1.7 | -1.6 | -0.1 | -0.2 | -0.4 | -0.3 | -0.3 | 0.0 | 0.1 | 0.0 |
| _0948 | *nagP* | -0.8 | -1.0 | -2.0 | -2.0 | -2.4 | -2.3 | -2.6 | -3.1 | 0.2 | 0.0 |
| _0946 | *nagA1* | -1.7 | -2.3 | -5.0 | -5.3 | -4.7 | -5.0 | -4.6 | -5.2 | 0.1 | 0.1 |
| _1198 | *nagA2* | -0.7 | -1.3 | 0.1 | 0.1 | -0.1 | -0.3 | 0.7 | 0.2 | -0.3 | -0.2 |

### (C) *Caulobacter crescentus* NA1000

| CCNA | | glucosamine (nitrogen) | | NAcGlc (carbon) | | | NAcGlc (nitrogen) | | glucose (carbon) | | |
|---|---|---|---|---|---|---|---|---|---|---|---|
| _00575 | *nagX* | -2.3 | -2.4 | -1.3 | -0.8 | -0.9 | -0.4 | -0.3 | -1.0 | -0.9 | -0.9 |
| _00452 | *nagA* | -1.4 | -1.5 | -2.3 | -1.4 | -3.4 | 0.3 | 0.2 | 0.3 | 0.4 | 0.3 |
| _00453 | *nagB* | -2.0 | -2.8 | -3.1 | -2.2 | -3.9 | 0.4 | -0.3 | -0.1 | 0.1 | -0.1 |

### (D) *Pedobacter sp.* GW460-11-11-14-LB5

| CA265 | | glucosamine (carbon) | | glucosamine (nitrogen) | NAcGln (carbon) | | glucose (carbon) | |
|---|---|---|---|---|---|---|---|---|
| _RS21920 | *nagX* | -0.8 | -0.7 | -2.3 | -0.3 | -0.2 | -0.0 | 0.2 |
| _RS08675 | *nagP* | -3.0 | -2.4 | -3.9 | -3.7 | -3.3 | -0.6 | -0.0 |
| _RS11300 | *nagK* | -2.0 | -2.2 | -1.7 | -3.1 | -2.9 | -0.2 | -0.0 |
| _RS21925 | *nagAB* | -3.5 | -4.9 | -4.3 | -5.1 | -4.6 | -0.2 | 0.1 |

### (F) Proposed role of NagX

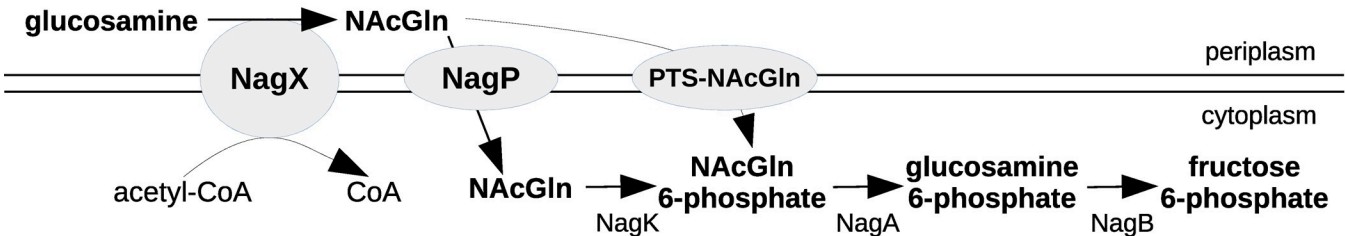

**Fig 3. Role of NagX in glucosamine utilization.** (A-E) Fitness data from five different bacteria with glucosamine or NAcGln as the sole source of carbon or nitrogen. As a control, we also show fitness with D,L-lactate or glucose as the carbon source. Each colored cell shows the fitness value for a gene in an individual experiment. The fitness of a gene is the log2 change in the relative abundance of mutants in that gene during 4–8 generations of growth (from inoculation at $OD_{600} = 0.02$ until saturation). Cells with strongly negative fitness are dark blue. (F) The proposed role of NagX.

deacetylase NagA was also involved in glucosamine utilization (Fig 3A–3D). And in four of the five bacteria, the transporter NagP or another putative sugar transporter were also involved in glucosamine utilization (Fig 3A, 3B, 3D and 3E).

NagX proteins are distantly related (25–31% amino acid identity) to human heparan-α-glucosaminide N-acetyltransferase (HGSNAT), which transfers acetyl groups from cytoplasmic acetyl-CoA to terminal glucosamine residues in lysosomal heparan sulfate [19]. Similarly, we propose that NagX is a transmembrane transacetylase that uses cytoplasmic acetyl-CoA to convert periplasmic glucosamine to N-acetylglucosamine (NAcGln). Although NagX is much shorter than HGSNAT, with 309–395 amino acids instead of 663, NagX contains the entire catalytic domain (PFam PF07786; [20]). Furthermore, the catalytic histidine which carries the acetyl group across the membrane is conserved: for instance, His72 of Shewana3_3111 aligns to His297 of HGSNAT (SwissProt Q68CP4). Once NAcGln is formed, it can be transported across the membrane and phosphorylated (such as by NagP and NagK, or by a phosphotransferase system), followed by deacetylation by NagA. Our proposal explains why NagA, NagP, and NagK are involved in glucosamine utilization as well as NAcGln utilization. Our proposal also explains why NagX is important for the utilization of glucosamine but not NAcGln (Fig 3A–3E, although NagX might be involved in NAcGln utilization in *Caulobacter crescentus*). We also noticed that in *Echinicola vietnamensis* KMM 6221, a putative acetyl-CoA synthase (*acs*) is important during glucosamine utilization (Fig 3E), but not in most other conditions (not shown); we speculate that it produces acetyl-CoA for NagX.

NagX is also distantly related to a putative N-acetylmuramate transporter (TfMurT) from *Tannerella forsythia* [21]. So we also considered that NagX might be a glucosamine transporter. However, this seems inconsistent with the involvement of the deacetylase NagA and of other sugar transporters in glucosamine utilization.

## Citrulline utilization via putative citrullinase CtlX

Using fitness data from *Phaeobacter inhibens* DSM 17395 (BS107), *Pseudomonas simiae* WCS417, and *Pseudomonas fluorescens* FW300-N2E3, we previously identified [4] a family of putative hydrolases that are involved in citrulline utilization (Fig 4A–4C). These hydrolases, which we will call CtlX, are distantly related to arginine deiminases, which hydrolyze arginine to citrulline and ammonia. We previously proposed that the arginine deiminase reaction might run in reverse [4]. But eQuilibrator estimates that the reverse reaction is thermodynamically unfavorable, with an equilibrium constant of under $10^{-6}$ $M^{-1}$ [22]. If arginine deiminase is operating in reverse, then the genes for converting citrulline to arginine (*argGH*) should be dispensable. We lack fitness data for *argGH* from *P. simiae* WCS417 or *P. inhibens* BS107, but in *P. fluorescens* FW300-N2E3, *argG* and *argH* were very important for fitness with citrulline as the sole source of either carbon or nitrogen (Fig 4A). Furthermore, the arginine deiminases and related enzymes that act on substrates with guanidino groups (-NH-C $(= NH_2^+)$-NH2) have two conserved substrate-binding aspartate residues [23], while CtlX has asparagines at these positions instead (FTR<u>D</u> → FPN<u>N</u> and HL<u>D</u> → HT<u>N</u>).

We noticed that CtlX is often encoded adjacent to ornithine cyclodeaminase *ocd* or ornithine/arginine N-succinyltransferase *aruG* (Fig 4D). These enzymes are also involved in citrulline utilization (Fig 4A–4C), which suggests that ornithine is an intermediate. This led us to consider that CtlX might hydrolyze citrulline to ornithine and carbamate (Fig 4E). The replacement of substrate-binding aspartates with asparagines seems consistent with an amide substrate.

Unfortunately, citrulline is not included in the IJSEM database [14], so we do not have a large data set of citrulline-utilizing bacteria. But *ctlX* is present in four of the five bacteria we have studied that grow with citrulline as the sole source of carbon. (Besides the three bacteria shown in Fig 4, *ctlX* is present in *P. fluorescens* FW300-N1B4, but we lack fitness data for the gene.) From a study of bacteria that can use citrulline as the sole source of carbon [24], we

**A** *Pseudomonas fluorescens* FW300-N2E3

| | citrulline (C / N) | | arginine (carbon) | | arginine (nitrogen) | | glucose (carbon) | |
|---|---|---|---|---|---|---|---|---|
| AO353_25635 *ctlX* | -4.7 | -5.8 | -0.1 | -0.1 | 0.1 | 0.0 | -0.1 | 0.1 |
| AO353_25630 *aruG* | -4.4 | -3.3 | -0.3 | -0.1 | -0.1 | 0.1 | -0.2 | 0.2 |
| AO353_03015 *aruF?* | -3.2 | -3.5 | -3.4 | -3.2 | -3.9 | -2.4 | -1.3 | -1.5 |
| AO353_04105 *argG* | -2.8 | -3.4 | -0.2 | -0.2 | 0.2 | -0.1 | -3.1 | -3.3 |
| AO353_09000 *argH* | -3.2 | -3.1 | -1.7 | -1.5 | -0.4 | -0.3 | -3.0 | -3.4 |

**B** *Pseudomonas simiae* WCS417

| | citrulline (carbon) | | arginine (carbon) | | glucose (carbon) | | |
|---|---|---|---|---|---|---|---|
| PS417_17580 *ctlX* | -4.8 | -3.5 | -0.4 | -0.1 | -0.0 | -0.1 | 0.2 |
| PS417_17585 *arcB* | -2.9 | -2.5 | 0.3 | 0.1 | 0.2 | 0.2 | -0.2 |

**C** *Phaeobacter inhibens* DSM 17395

| | citrulline (carbon) | | citrulline (nitrogen) | | glucose (carbon) | | | |
|---|---|---|---|---|---|---|---|---|
| PGA1_c16380 *citX* | -4.2 | -4.8 | -2.0 | -2.3 | -0.4 | -0.5 | -0.3 | -0.1 |
| PGA1_c16390 *arcB* | -5.1 | -3.4 | -1.6 | -3.2 | 0.2 | -0.3 | 0.4 | -0.1 |

**D**

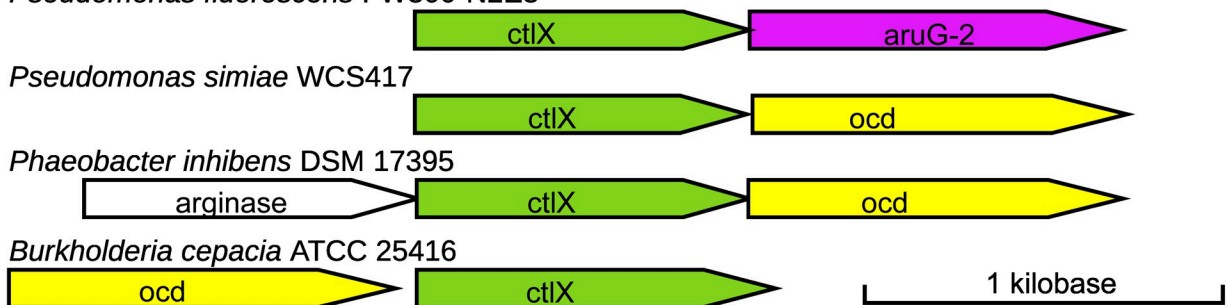

**E**

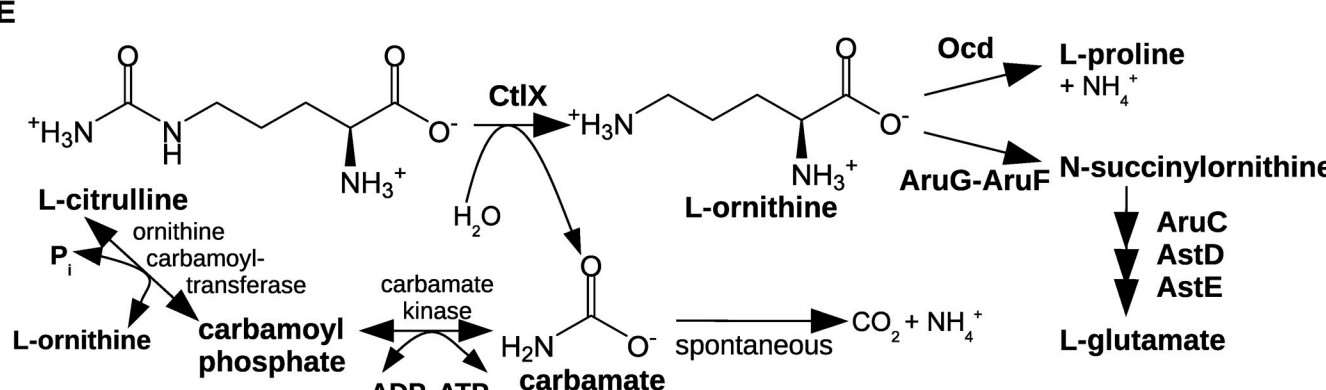

**Fig 4. Putative citrullinase CtlX.** (A-C) In diverse bacteria, *ctlX* and either ornithine cyclodeaminase (*ocd*) or ornithine/arginine succinyltransferase (*aruFG*) are important for the utilization of citrulline as a carbon source. The color-coded cells show fitness values, which are log2 changes in the relative abundance of mutants in each gene. (D) Gene neighborhoods of *ctlX*. The drawing is modified from Gene Graphics [25]. (E) Pathways of citrulline utilization.

found two with genome sequences, and both encode *ctlX* (C8E02_RS07400 from *Vogesella indigofera* ATCC 19706 = DSM 3303, and DM41_RS32400 from *Burkholderia cepacia* NCTC 10743 = ATCC 25416 = DSM 7288). Furthemore, *ctlX* from *B. cepacia* is encoded adjacent to *ocd* (Fig 4D). So CtlX is widespread in citrulline-utilizing bacteria.

To further investigate the role of CtlX, we collected additional fitness data for *P. fluorescens* FW300-N2E3, *P. simiae* WCS417, and *P. inhibens* DSM 17395 during growth with varying concentrations of citrulline or ornithine as the sole source of carbon. We had expected that CtlX would be important for the utilization of citrulline, but not ornithine. Instead, we observed that CtlX was important for the utilization of both citrulline and ornithine in all three bacteria. We suspect that ornithine is being converted to citrulline and then arginine by enzymes of the arginine biosynthesis pathway, and that CtlX is important for fitness because it counteracts this. First, *P. fluorescens* FW300-N2E3 has three ways to consume arginine: the arginine succinyltransferase pathway, the arginine decarboxylase pathway, and arginine deiminase ArcA, which converts arginine to citrulline. Genes from all three pathways are strongly detrimental to fitness during growth on ornithine; in other words, mutants in these pathways are enriched after growth on ornithine (S1 Fig). This suggests that an excess of arginine is being formed (although we do not understand why disrupting just one of three catabolic pathways is beneficial). Second, in *P. simiae* WCS417, several genes from the arginine succinyltransferase pathway are important for fitness during growth on ornithine (S2 Fig). This is consistent with flux to arginine in excess of requirements for protein synthesis, although these genes could be involved in ornithine catabolism instead, as AruFG can succinylate both arginine and ornithine [26]. We also noticed that all four transposon insertions within the *ctlX* of *P. simiae* WCS147 have the antibiotic resistance marker in the antisense orientation, which might prevent expression of the downstream ornithine cyclodeaminase (*ocd*) in these strains. *Ocd* is important for utilization of ornithine (S2 Fig), so the phenotype of insertions in *ctlX* could be a polar effect. Third, in *P. inhibens* DSM 17395, arginase (which hydrolyzes arginine to ornithine and urea) was very important for fitness during growth on either ornithine or citrulline, which again implies excess flux to arginine (S3 Fig). Because of the complexity of citrulline and arginine metabolism, biochemical studies will be needed to prove the function of CtlX. In the current release of GapMind, we assume that CtlX converts citrulline to ornithine.

The only citrullinase from bacteria that has been reported before, Ctu from *Francisella tularensis* [27], is not homologous to CltX (PFam PF00795, not PF02274). Also, many *Pseudomonas* can use ornithine carbamoyltransferase and carbamate kinase (both in reverse) to consume citrulline and form ATP (Fig 4E). (Both of the *Pseudomonas* with the putative citrullinase also encode carbamate kinase, but *Phaeobacter inhibens* DSM 17395 does not.) In *Pseudomonas aeruginosa*, these enzymes are repressed under aerobic conditions [28], and all of our experiments with citrulline were conducted aerobically, so the carbamate kinase pathway may not have been expressed. Although the carbamate kinase pathway generates one more ATP per molecule of citrulline than the citrullinase pathway, the first step of the carbamate kinase pathway (ornithine carbamoyltransferase in reverse) is thermodynamically quite unfavorable, with an estimated equilibrium constant of $5 \cdot 10^{-6}$ [22]. So we speculate that the citrullinase pathway is faster, which would explain why it is preferred when oxygen is available.

## An alternative 2-deoxy-5-keto-D-gluconate 6-phosphate aldolase for myo-inositol utilization

2-deoxy-5-keto-D-gluconate 6-phosphate aldolase (EC 4.1.2.29) is involved in myo-inositol catabolism via inosose dehydratase and 5-deoxy-D-glucuronate. As far as we know, the only

previously-characterized enzymes are IolJ from *Bacillus subtilis* [29] and a similar protein from *Phaeobacter inhibens*, PGA1_c07220, which was identified using fitness data [4]. Of the 11 bacteria for which we have fitness data with myo-inositol as the sole carbon source, just two encode IolJ-like proteins, so we searched for alternative aldolases using the fitness data. We noticed that in the other nine bacteria, a putative 2-deoxy-5-keto-D-gluconate kinase (IolC) is fused to an uncharacterized domain, DUF2090 (PFam PF09863). All of these fusion proteins were important for fitness during myo-inositol utilization but not in most other conditions (Fig 5).

DUF2090 is related to aldolases: for instance, D-tagatose-bisphosphate aldolase LacD from *Streptococcus pyogenes* (PDB:5ff7) has a statistically significant alignment to PF09863.9 (uncorrected E = $6.5 \cdot 10^{-8}$, hmmsearch 3.3.1). The catalytic residues of LacD are Lys126 and Glu164 [30]. When we aligned LacD and the DUF2090 fusion proteins (via the PFam model and hmmsearch), we found that these catalytic residues were fully conserved. For instance, BPHYT_RS13910 from *Burkholderia phytofirmans* PsJN has Lys493 and Glu531. We propose that DUF2090 is the missing 2-deoxy-5-keto-D-gluconate 6-phosphate aldolase.

When we examined the genomes of diverse myo-inositol-utilizing microbes from the IJSEM database [14], we found that none contained IolJ, but 7 of 15 (47%) contained DUF2090, and in each case, DUF2090 was fused to IolC. Just 22 of 232 genomes (9%) from organisms not known to utilize myo-inositol contained DUF2090, which was significantly less (odds ratio 0.12, P = 0.0005, Fisher exact test). (To identify members of DUF2090, we used hmmsearch with PF09863.9 and the trusted cutoff, and proteins that had higher bit scores for alignments to the DeoC/LacD family (PF01791.9) than to PF09863.9 were ignored.) If we combine the 11 myo-inositol-utilizing bacteria with fitness data with the 15 microbes from IJSEM, then of the 26 genomes, 16 encode IolC-DUF2090 and just 2 encode IolJ. Thus, DUF2090 is associated with myo-inositol utilization, which supports our prediction that DUF2090 domains are 2-deoxy-5-keto-D-gluconate 6-phosphate aldolases.

## Lactose utilization via a putative periplasmic 3'-ketolactose hydrolase

In *Caulobacter crescentus*, lactose is thought to be consumed via oxidation to 3'-ketolactose and hydrolysis to glucose and 3-ketogalactose [31]. The lactose 3-dehydrogenase has three known components, which are encoded by *lacABC*, and all three components are required for lactose utilization [31]. As far as we know, there is no experimental evidence for 3'-ketolactose hydrolysis by *C. crescentus*, nor has this activity been linked to sequence. But a 3'-ketolactose hydrolase was partially purified from *Agrobacterium tumefaciens*, which also contains lactose 3-dehydrogenase [32]. The enzyme from *A. tumefaciens* produced glucose; the other product could not be determined, but it is expected to be 3-ketogalactose [31].

We found that in *C. crescentus* NA1000, CCNA_01705 is important for lactose utilization (Fig 6A). Across 198 fitness experiments with diverse growth conditions, including 14 carbon sources, we identified a strong defect for mutants in CCNA_01705 (gene fitness of -2 or less) only during growth on lactose (3/3 replicates) or on the trisaccharide raffinose (1/3 replicates; Fig 6A). CCNA_01705 is encoded near the lactose 3-dehydrogenase (Fig 6B) and contains a single DUF1080 domain (PF06439). The only characterized proteins with this domain architecture that we are aware of are the 3-ketotrehalose hydrolase BT2157 [8] and the endo-xanthanase/lichenase THTE_1561 [33]. Since 3-ketotrehalose and 3'-ketolactose are similar compounds, we propose that CCNA_01705 is the 3'-ketolactose hydrolase of *C. crescentus*.

CCNA_01705 has a putative signal peptide [34] and we propose that it is located in the periplasm. LacABC is membrane bound and oxidizes lactose in the periplasm [31], so 3'-ketolactose would be hydrolyzed there as well.

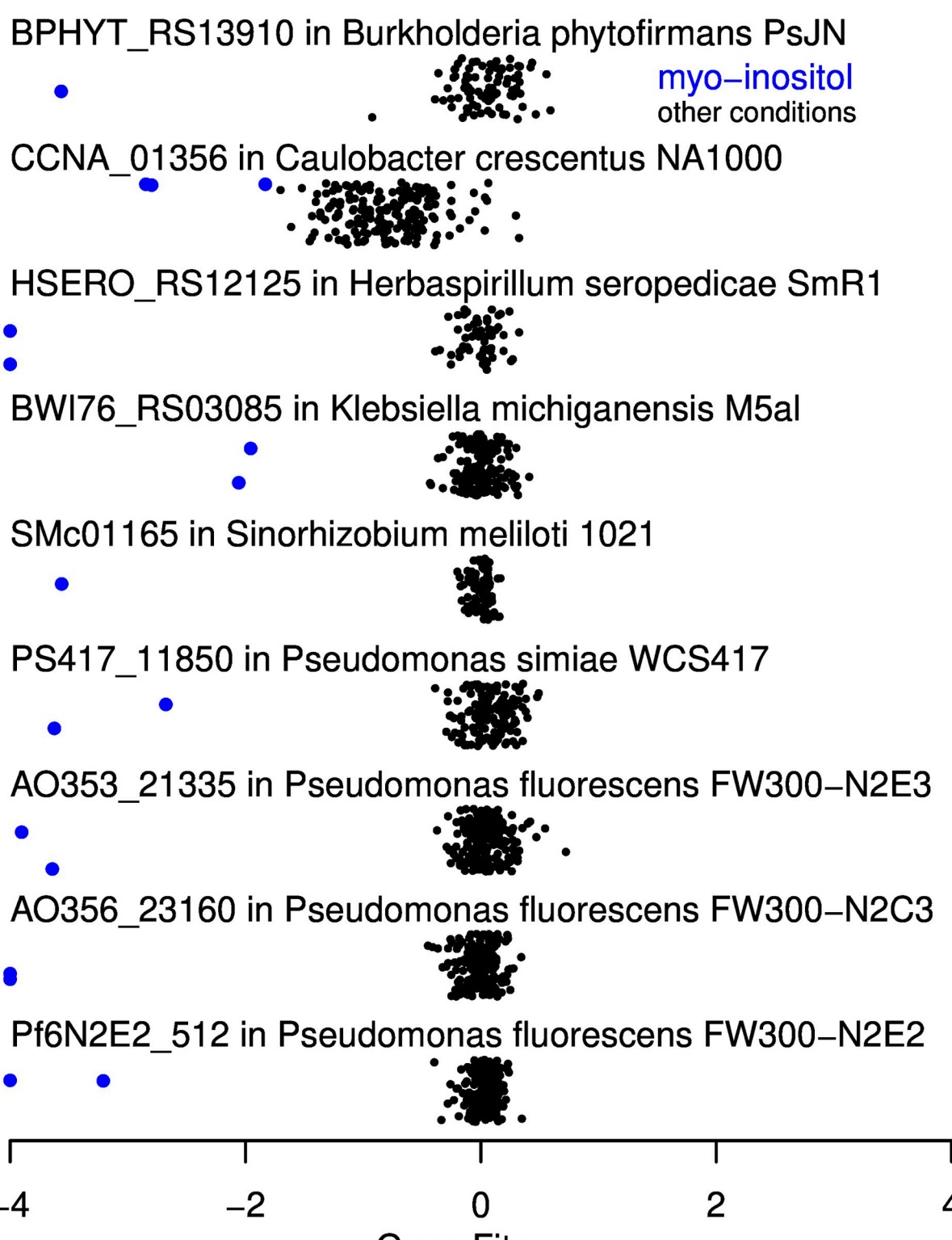

**Fig 5. IolC-DUF2090 fusion proteins are important for myo-inositol utilization.** Each point shows a gene fitness value (*x* axis) from a separate experiment. Values under -4 are shown at -4. The *y* axis is arbitrary. Experiments with myo-inositol as the sole source of carbon are highlighted.

**A    Mutant fitness data for *Caulobacter crescentus* NA1000**

| | | lactose | | | raffinose | | | salicin | | | glucose | | |
|---|---|---|---|---|---|---|---|---|---|---|---|---|---|
| CCNA_01698 | DUF1080 protein | 0.3 | 0.2 | 0.0 | -2.2 | -2.3 | -1.3 | -0.3 | -0.3 | -0.4 | -0.3 | -0.2 | -0.1 |
| CCNA_01699 | LacI-type regulator | 2.2 | 2.0 | 2.2 | 1.3 | 1.3 | 1.2 | 1.1 | 1.0 | 1.2 | 0.3 | 0.4 | 0.2 |
| CCNA_01700 | 3-ketohexose transporter? | -3.0 | -2.3 | -2.7 | -2.6 | -2.6 | -1.7 | -2.9 | -1.3 | -2.9 | -1.6 | -1.3 | -1.3 |
| CCNA_01701 | sugar epimerase? | -1.8 | -2.0 | -1.8 | -1.4 | -0.9 | -1.1 | -1.5 | -1.1 | -2.5 | -0.7 | -0.2 | -0.7 |
| CCNA_01702 | 3-ketohexose reductase? | -1.7 | -1.3 | -1.8 | -1.4 | -1.2 | -0.8 | -1.5 | -0.9 | -1.8 | -0.3 | 0.0 | -0.1 |
| CCNA_01703 | sugar epimerase? | -2.0 | -2.1 | -2.1 | -1.5 | -1.4 | -1.1 | -1.8 | -0.9 | -2.1 | 0.1 | 0.1 | 0.2 |
| CCNA_01704 | **lactose dehydrogenase, lacB** | -1.8 | -1.7 | -3.3 | -2.1 | -1.3 | -1.5 | -1.1 | -0.9 | -2.0 | -0.5 | -0.2 | 0.1 |
| CCNA_01705 | **3'-ketolactose hydrolase (DUF1080)** | -2.5 | -2.5 | -2.6 | -2.0 | -1.8 | -1.6 | -1.2 | -0.6 | -1.1 | -1.2 | -0.9 | -0.9 |
| CCNA_01706 | **lactose dehydrogenase, lacA** | -1.7 | -2.3 | -2.1 | -2.3 | -1.8 | -1.3 | -1.3 | -0.9 | -2.1 | -0.6 | -0.4 | -0.5 |
| CCNA_01707 | **lactose dehydrogenase, lacC** | -2.4 | -1.5 | -2.2 | -2.1 | -2.3 | -0.5 | -2.6 | -1.4 | -1.3 | 0.3 | -0.0 | 0.3 |
| CCNA_01159 | glucose transporter | -1.9 | -2.0 | -2.0 | -1.0 | -0.9 | -0.6 | -0.9 | -0.9 | -1.0 | -3.7 | -4.2 | -4.1 |
| CCNA_02136 | glucose-6-phosphate dehydrogenase | | | | | | | | | | | | |
| CCNA_02135 | 6-phosphogluconolactonase | -1.7 | -2.2 | -2.0 | -1.8 | -1.4 | -1.0 | -1.4 | -1.2 | -2.0 | -4.8 | -3.9 | -4.3 |
| CCNA_02134 | phosphogluconate dehydratase | -2.3 | -2.2 | -2.0 | -0.6 | -0.5 | -0.4 | -1.5 | -1.4 | -1.4 | -4.6 | -4.4 | -4.5 |
| CCNA_02133 | glucokinase | -2.4 | -2.0 | -2.3 | 1.6 | 1.7 | 1.4 | -1.6 | -1.7 | -1.6 | -4.2 | -4.1 | -4.7 |
| CCNA_00830 | β-galactosidase | -1.2 | -1.4 | -1.4 | -1.0 | -1.3 | -0.8 | -1.1 | -0.7 | -1.0 | -0.9 | -0.8 | -1.0 |

**B**

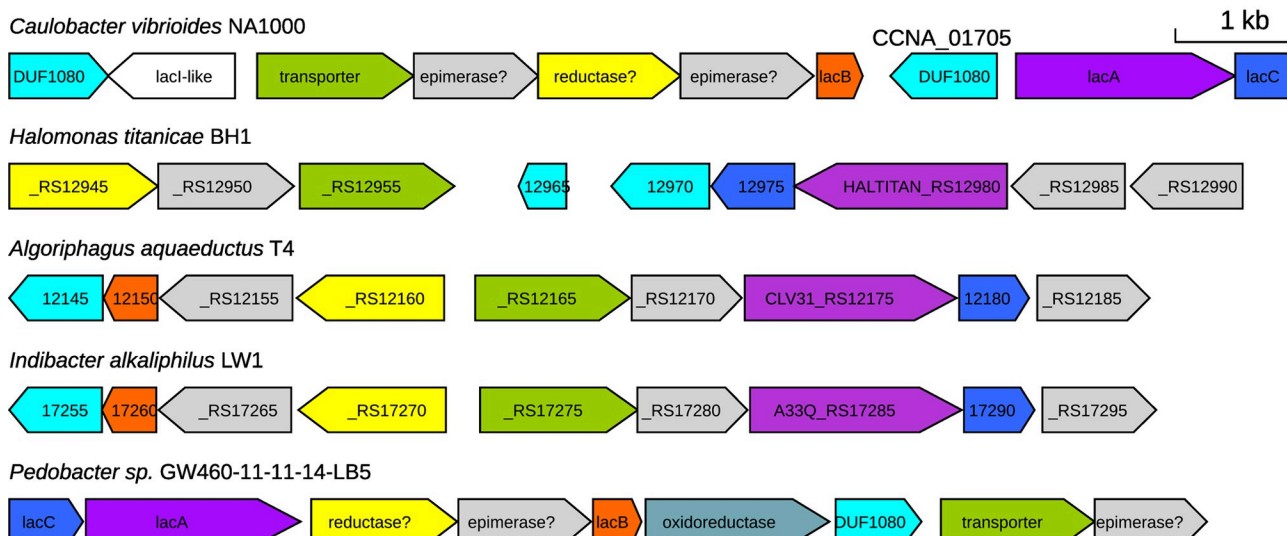

**C    Mutant fitness data for *Pedobacter sp.* GW460-11-11-14-LB5**

| CA265_ | | lactose | trehalose | | | | | | | lactulose | | salicin | | cellobiose | | lactitol | | maltitol | | methyl β-galactose | | glucose | |
|---|---|---|---|---|---|---|---|---|---|---|---|---|---|---|---|---|---|---|---|---|---|---|---|
| _RS15340 | **lacC** | -4.5 | -5.1 | -5.8 | -3.2 | -2.9 | -5.8 | -7.3 | -4.0 | -7.7 | -6.6 | -5.1 | -4.1 | -4.5 | -3.4 | -4.0 | -4.7 | -3.8 | -3.8 | -3.6 | 0.1 | 0.3 |
| _RS15345 | **lacA** | -4.1 | -5.1 | -5.5 | -4.4 | -3.8 | -6.3 | -5.6 | -6.1 | -4.2 | -5.1 | -5.1 | -3.6 | -3.7 | -4.3 | -4.9 | -4.6 | -4.3 | -3.7 | -3.6 | 0.1 | 0.1 |
| _RS15350 | reductase? | -3.4 | -4.0 | -3.8 | -3.3 | -2.4 | -5.3 | -6.9 | -6.5 | -4.8 | -6.6 | -5.5 | -2.6 | -2.8 | -3.8 | -3.5 | -4.2 | -3.8 | -4.2 | -4.0 | -0.1 | 0.1 |
| _RS15355 | epimerase? | -2.8 | -3.6 | -3.5 | -4.3 | -2.5 | -4.9 | -4.2 | -7.1 | -3.7 | -4.1 | -3.7 | -3.5 | -2.8 | -2.9 | -3.6 | -3.5 | -3.1 | -3.5 | -3.4 | -0.2 | 0.0 |
| _RS15360 | **lacB** | | | | | | | | | | | | | | | | | | | | | |
| _RS15365 | oxidoreduc. | -0.6 | -0.1 | -0.0 | -0.2 | -0.2 | -0.2 | -0.4 | -1.0 | -0.7 | -1.8 | -1.8 | -0.8 | -0.8 | -1.1 | -0.8 | -0.4 | -0.3 | -0.2 | -0.3 | -0.2 | -0.4 |
| _RS15370 | DUF1080 | -0.4 | -0.1 | -0.2 | -0.1 | -0.0 | -0.0 | -0.2 | 0.4 | 0.7 | -1.1 | -0.7 | -0.1 | -0.4 | 0.5 | 0.2 | -0.0 | 0.1 | 2.8 | 2.9 | 0.4 | 0.5 |
| _RS15375 | transporter | -1.7 | -2.9 | -3.4 | -2.6 | -2.4 | -5.1 | -4.0 | -4.2 | -4.5 | -4.6 | -5.3 | -1.6 | -1.6 | -4.2 | -3.6 | -3.9 | -3.2 | -3.9 | -4.1 | -0.2 | -0.1 |
| _RS15380 | epimerase? | -0.3 | 0.2 | 0.3 | -0.3 | -0.1 | -0.2 | 0.3 | -1.4 | -2.4 | -0.3 | -0.3 | -0.3 | -0.4 | -2.5 | -2.1 | -0.1 | -0.0 | -1.0 | -1.2 | -0.4 | 0.2 |

**Fig 6. A putative 3'-ketolactose hydrolase from the DUF1080 family is involved in lactose utilization.** (A) Fitness data for *Caulobacter crescentus* NA1000 grown in different carbon sources (data from [4]). (B) LacABC-type dehydrogenases are encoded near DUF1080 in diverse lactose-utilizing bacteria. (C) Fitness data from *Pedobacter sp.* GW460-11-11-14-LB5 grown in different carbon sources (data from [4,8]). Missing fitness values are shown in grey.

The lactose dehydrogenase of *C. crescentus* is also reported to act on salicin (a phenolic β-glucoside), and *lacABC* are required for salicin utilization in some genetic backgrounds [31]. Consistent with this, in our fitness data, *lacABC* is important for growth on salicin (Fig 6A). *LacABC* was also important for growth on raffinose, a trisaccharide (Fig 6A). CCNA_10705 had a milder phenotype on raffinose or salicin than on lactose (Fig 6A), but another DUF1080 protein is encoded nearby (CCNA_01698) and is important for utilization of raffinose (Fig 6A). The two DUF1080 proteins could be genetically redundant during utilization of salicin (presumably acting on 3-ketosalicin). A 3-ketoglycoside hydrolase from *Agrobacterium tumefaciens* acts on a variety of 3-ketoglucosides [35], so we suspect that CCNA_10705 is active on 3-ketosalicin and some other 3-ketoglycosides as well as on 3'-ketolactose.

What is the fate of the putative products of 3'-ketolactose hydrolysis, glucose and 3-ketogalactose? Glucose is probably taken up by a transporter (CCNA_01159) and consumed by the Entner-Doudoroff pathway (CCNA_02136-CCNA_02133); these glucose utilization genes are important during growth on lactose (Fig 6A). Also, upstream of *lacB* are a transporter, two sugar epimerases and a sugar reductase (Fig 6B) that are important for lactose utilization (Fig 6A); these genes could be involved in the utilization of 3-ketogalactose, possibly by reduction to a hexose. In fact, there is biochemical evidence for a 3-ketoglucose reductase in *A. tumefaciens* [36].

There are two well-described pathways for lactose utilization: lactose hydrolase (β-galactosidase); or phosphorylation to lactose 6'-phosphate and hydrolysis by a phospho-β-galactosidase [7]. *C. crescentus* does have β-galactosidase activity [31], and it encodes a putative β-galactosidase, CCNA_00830, which is 60% identical to a characterized β-galactosidase from *Xanthomonas campestris* [37]. Mutants of CCNA_00830 gene were only mildly reduced in abundance after growth in lactose, and had similar phenotypes during growth in other carbon sources (Fig 6A). The β-galactosidase pathway may occur in parallel with the lactose oxidation pathway. Alternatively, as lactose oxidation is required for the induction of β-galactosidase expression [31], utilization could occur primarily via β-galactosidase, and the mild phenotype for CCNA_00830 could be due to genetic redundancy (there are two other medium-confidence candidates for β-galactosidase). In this hypothetical scenario, the expression of both genetically-redundant β-galactosidase genes must depend on lactose oxidation, so we consider it unlikely.

A putative lactose dehydrogenase from *Pedobacter sp*. GW460-11-11-14-LB5 is also important for the utilization of lactose, salicin, and several other glycosides (Fig 6C). This strain encodes ten DUF1080 proteins and at least five putative β-galactosidases, but we did not identify phenotypes for any of the DUF1080 or β-galactosidase genes with lactose as the carbon source (all fitness values were between -0.4 and +0.2). This could be due to genetic redundancy.

Among the microbes in the IJSEM database, we found that the presence of DUF1080 in the genome is associated with lactose utilization: DUF1080 is present in 40% of lactose-utilizing microbes but only 14% of other microbes (odds ratio = 4.1, P = $5.2 \cdot 10^{-5}$, Fisher exact test). (DUF1080 proteins were identified using the trusted cutoff for PF06439.11.) Of the 57 lactose-utilizing genomes from the IJSEM database, 14 appear to encode neither β-galactosidase nor phospho-β-galactosidase. (No high- or medium-confidence candidates were identified by GapMind.) Of these 14 genomes, four encode proteins similar to LacA (40% identity and above) and DUF1080, and in three of these genomes, the LacA and DUF1080 proteins are encoded near each other, along with other proteins that are similar to the gene cluster from *C. crescentus* (Fig 6B). A caveat is that two of these bacteria (*Halomonas titanicae* BH1 and *Algoriphagus aquaeductus* T4) were reported to have β-galactosidase activity [38,39]; however, the third, *Indibacter alkaliphilus* LW1, is β-galactosidase negative [40].

These results suggest that LacABC and DUF1080 proteins function together in the utilization of lactose by diverse bacteria. (*C. crescentus* is an α-Proteobacterium, while *Pedobacter* and *Indibacter* are Bacteroidetes.) Although the fate of the 3-ketogalactose is unknown, in the current release of GapMind, we assume that LacABC and DUF1080 suffice to release periplasmic glucose, which can then be consumed.

## An alternative oxepin-CoA hydrolase for phenylacetate utilization

Phenylacetate is an end product of phenylalanine fermentation, and phenylacetate or phenylacetyl-CoA are common intermediates in the degradation of phenylalanine and other aromatic compounds. The aerobic pathway for phenylacetate utilization [41,42] begins by activation to phenylacetyl-CoA, oxygenation to 1,2-epoxyphenylacetyl-CoA, isomerization to oxepin-CoA, hydrolytic ring-opening to 3-oxo-5,6-didehydrosuberyl-CoA semialdehyde, and oxidation to 3-oxo-5,6-didehydrosuberyl-CoA. Additional thiolase, isomerase, dehydrogenase, and enoyl-CoA hydratase enzymes convert this to acetyl-CoA and succinyl-CoA (Fig 7A). In *E. coli*, the ring opening reaction and the next step in the pathway, the oxidation of 3-oxo-5,6-didehydrosuberyl-CoA semialdehyde, are catalyzed by PaaZ, which combines an enoyl-CoA hydratase (ECH) domain that performs ring opening with an aldehyde dehydrogenase domain [43]. But in many other bacteria that encode this pathway, the 3-oxo-5,6-didehydrosuberyl-CoA semialdehyde dehydrogenase is a separate protein (for instance, PacL, [43]). To our knowledge, the oxepin-CoA hydrolase from these bacteria has not been identified. Teufel and colleagues did identify a protein (ECH-Aa) that had some activity as an oxepin-CoA hydrolase, but ECH-Aa was ~1,000 times more active as a crotonyl-CoA hydratase than as oxepin-CoA hydrolase, so it is not clear if ECH-Aa's oxepin-CoA hydrolase activity is physiologically relevant [43].

To study this question, we analyzed fitness data from *Paraburkholderia bryophila* 376MFSha3.1 with phenylacetate as the carbon source (Robin Herbert and Trenton Owens, personal communication). Most of the genes of the aerobic pathway were identified in the genome and were important for phenylacetate utilization, including the phenylacetate-CoA ligase *paaK*, the oxygenase *paaABCDE*, the isomerase *paaG*, a *pacL*-like 3-oxo-5,6-didehydrosuberyl-CoA semialdehyde dehydrogenase, the thiolase *paaJ*, and the enoyl-CoA hydratase *paaF* (Fig 7B). The only missing steps were the oxepin-CoA hydrolase and the 3-hydroxyadipoyl-CoA dehydrogenase (PaaH). Using the fitness data, we identified candidates for both steps.

First, a putative enoyl-CoA hydratase, H281DRAFT_04594 was important for phenylacetate utilization (Fig 7B). A closely related protein from *Burkholderia sp*. OAS925 (97% identity) is also important for phenylalanine utilization (Ga0395975_5191, fitness = -4.1 and -3.9, Marta Torres, personal communication), which confirms our genetic data. We predict that these proteins provide the missing oxepin-CoA hydrolase activity. H281DRAFT_04594 is related to enoyl-CoA hydratases that form (3S)-hydroxyacyl-CoA from 2-*trans*-enoyl-CoA, while the ECH domain of PaaZ is related to enoyl-CoA hydratases that form (3R)-hydroxyacyl-CoA. Both families of hydratases use acid-base chemistry to act on CoA thioesters, and neither oxepin-CoA nor the hydrolysis product have chiral centers (except within the coenzyme A group), so either type of ECH domain could catalyze the hydrolysis of oxepin-CoA. H281DRAFT_04594 is 32% identical to enoyl-CoA hydratase from rat liver, whose catalytic mechanism has been studied [44]. The side chains that participate in catalysis (E144 and Q162) are not conserved in H281DRAFT_04594: the corresponding residues are S118 and M135, respectively. This suggests that H281DRAFT_04594 has another function, which is consistent with our proposal.

## A. Aerobic phenylacetate degradation

## B. Fitness data from *Paraburkholderia bryophila* 376MFSha3.1

| Gene | Description | 10 mM phenylacetate | | 10 mM D-glucose | | 20 mM D-glucose | | | |
|------|-------------|------|------|------|------|------|------|------|------|
| H281DRAFT_05720 | paaK | -5.7 | -3.5 | -0.0 | -0.0 | 0.0 | 0.2 | 0.2 | 0.2 |
| H281DRAFT_05858 | paaA | -4.3 | -3.2 | 0.0 | -0.3 | -0.1 | -0.1 | -0.2 | 0.0 |
| H281DRAFT_05857 | paaB | -3.0 | -2.3 | -0.2 | -0.1 | -0.2 | 0.1 | -0.4 | -0.3 |
| H281DRAFT_05856 | paaC | -5.0 | -4.1 | -0.4 | -0.1 | -0.3 | 0.0 | -0.2 | -0.1 |
| H281DRAFT_05855 | paaD | -5.4 | -4.0 | 0.1 | -0.5 | 0.9 | -0.1 | -1.0 | -0.1 |
| H281DRAFT_05854 | paaE | -5.7 | -4.0 | -0.0 | -0.1 | -0.2 | 0.1 | 0.2 | -0.5 |
| H281DRAFT_05722 | paaG | -6.0 | -3.0 | -0.1 | -0.1 | 0.2 | -0.1 | 0.2 | 0.2 |
| H281DRAFT_04594 | putative oxepin-CoA hydrolase | -5.4 | -3.2 | 0.0 | 0.1 | -0.2 | 0.1 | 0.0 | -0.1 |
| H281DRAFT_05724 | pacL | -4.7 | -2.8 | 0.0 | -0.2 | -0.2 | -0.2 | -0.0 | 0.2 |
| H281DRAFT_05723 | paaJ | -5.2 | -3.6 | 0.4 | 0.2 | 0.0 | 0.1 | 0.1 | -0.2 |
| H281DRAFT_05725 | paaF | -5.8 | -3.0 | 0.1 | 0.2 | 0.2 | 0.2 | 0.0 | -0.0 |
| H281DRAFT_00361 | 3-hydroxyacyl-CoA dehydrogenase / ECH | -1.8 | -2.1 | -0.1 | 0.3 | 0.2 | 0.3 | -0.6 | -0.4 |

**Fig 7. Phenylacetate utilization via an alternative oxepin-CoA hydrolase.** (A) The aerobic pathway for phenylacetate utilization. (B) Fitness data from *P. bryophila* 376MFSha3.1 growing in minimal media with phenylacetate or glucose as the carbon source. Except for the experiments with 20 mM glucose, the media also contained 1% dimethylsulfoxide (by volume).

Second, the gene for the 3-hydroxyadipoyl-CoA dehydrogenase PaaH was not clearly identified, but there are at least three 3-hydroxyacyl-CoA dehydrogenases that might have this activity. One of them, H281DRAFT_00361, was important for phenylacetate utilization (Fig 7B). A close homolog from *B. phytofirmans* PsjN was also important for phenylacetate utilization (BPHYT_RS13545, fitness = -1.7 or -2.0; data from [45]). H281DRAFT_00361 is 49% identical to PimB from *Rhodopseudomonas palustris*; the *pim* operon is involved in dicarboxylic fatty acid degradation [46], which suggests that PimB may be active on 3-hydroxyadipoyl-CoA (the 3-hydroxyacyl-CoA intermediate in adipate degradation). H281DRAFT_00361 has an ECH domain as well as an aldehyde dehydrogenase domain; we do not have a proposal for the role of its ECH domain.

## Annotation of 299 diverged enzymes and transporters

While developing GapMind, we used the fitness data to identify transporters and enzymes that were important for utilization of various carbon sources, and hence to predict these proteins' functions. Overall, we annotated 716 proteins, comprising 555 enzymes and 161 transporters or transporter components. (Proteins whose functions we had previously identified from the fitness data are not included in these counts.) Many of these proteins are distantly related to previously-characterized proteins from the seven curated databases that GapMind relies on (Fig 8A). For proteins that were over 40% identical to one or more characterized proteins, 22% (117 of 534) had a different function than their best hit. For example, PS417_22145 from *Pseudomonas simiae* WCS417 is 88% identical to GtsA from *P. putida* KT2440, which is reported in the transporter classification database (TCDB) to be the substrate-binding component of a

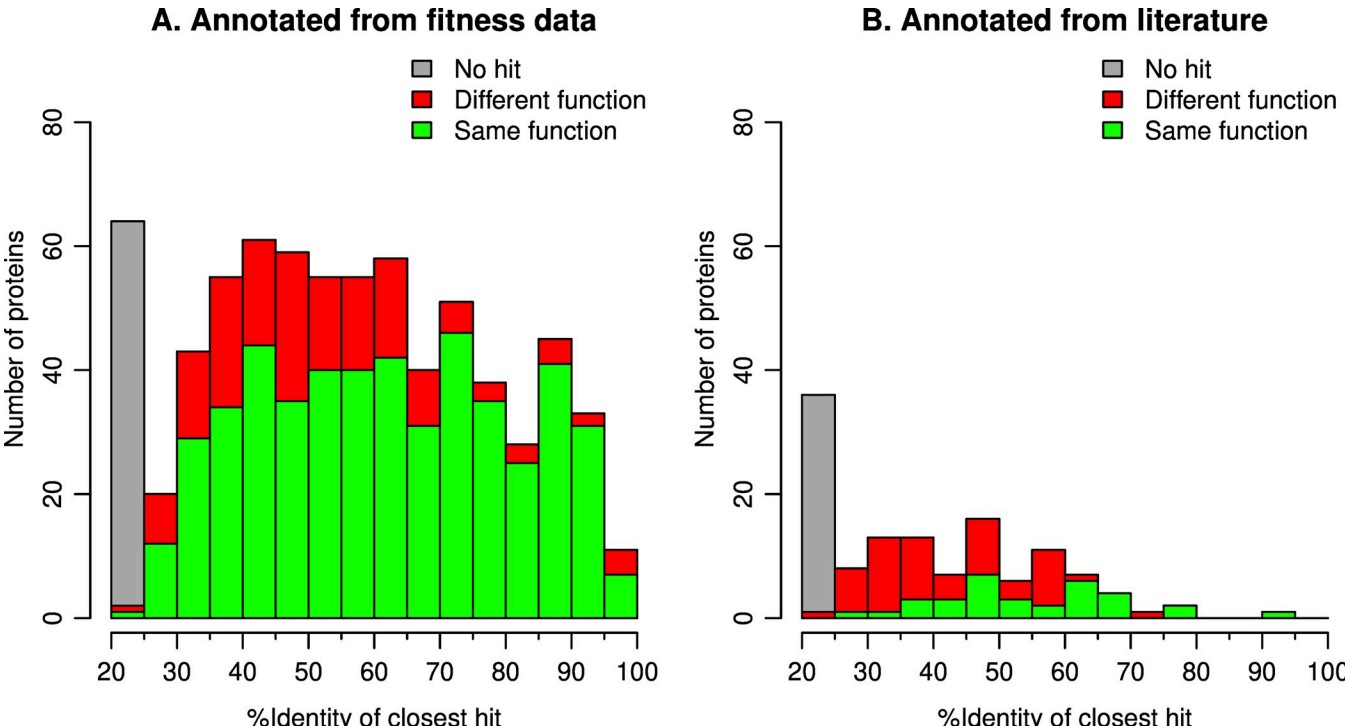

**Fig 8. Similarity of the proteins that we annotated to previously-characterized proteins from seven curated databases.** Panel A shows the 716 proteins that we annotated using fitness data, and panel B shows the 125 proteins that we annotated using the scientific literature. Homologs were identified using protein BLAST against a database of 125,685 experimentally-characterized proteins. We required E < 0.001 and 70% coverage of both the query and the subject. Proteins whose functions we had previously identified using the fitness data were not included in the database.

glucose transporter. PS417_22145 was important for the utilization of D-glucose 6-phosphate (fitness = -3.2 and -2.0) and D-xylose (fitness = -1.8 and -1.6) but not in most other conditions (data of [4,45]; also, the other components of this ABC transporter had similar phenotypes). Glucose 6-phosphate may be hydrolyzed to glucose before uptake, which would explain why a glucose transporter is important for fitness; but the phenotype during growth on D-xylose suggests that PS417_22145 binds xylose as well as glucose. Indeed, in strains of *P. putida* that were engineered to utilize xylose, GtsA is required for xylose utilization [47]. This information is not in TCDB: since the xylose-utilizing strains of *P. putida* had mutations in GtsA, it is not clear if the wild-type protein from *P. putida* binds to xylose. But GtsA from *Pseudomonas simiae* WCS417 does seem to be involved in xylose transport. Overall, we used the fitness data to identify functions for 299 diverged proteins that have a different function than their closest characterized homolog or are less than 40% identical to any characterized protein in the databases.

## Curation of enzymes and transporters from the literature

While developing GapMind, we identified 125 proteins that have published experimental data about their function, are relevant to the utilization of the 62 carbon sources, but are not included in any of the curated databases. For example, in *Pseudomonas putida* KT2440, the putative lactonase PP_1170 is important during growth on D-glucuronate and D-galacturonate (fitness < -2, Mitchell Thompson and Matthias Schmidt, personal communication), but not in over 100 other experiments (all fitness ≥ -0.5). A uronate dehydrogenase (PP_1171) is also important for glucuronate utilization, which indicates that *P. putida* uses an oxidative pathway and suggests that PP_1170 is a glucurono-1,5-lactonase. This reaction is not linked to protein sequences by any of the curated databases we used, so at first we thought we had identified a novel enzyme. But by using PaperBLAST [9], we found that PP_1170 is 72% identical to PSPTO_1052, which hydrolyzes D-glucurono-1,5-lactone *in vitro* [48]. GapMind now associates the glucurono-1,5-lactonase reaction with PP_1170, PSPTO_1052, and five other lactonases studied by [48].

Of the 125 proteins we curated from the literature, 61 are enzymes and 64 are transporters. The majority of these proteins are quite diverged from characterized proteins in the databases, or have different functions (Fig 8B). The median similarity to the most-similar characterized protein is 38%.

## Quality of GapMind's results

To assess the quality of GapMind's results, we examined its predictions for organisms that are reported to grow, or not, with these compounds as the sole source of carbon. First, we compared GapMind's results to growth data for 29 heterotrophic bacteria across 57 of the 62 carbon sources in GapMind [4,8]. (Deoxyinosine, deoxyribonate, mannitol, phenylacetate and sucrose were not included because we do not have comprehensive growth data.) As shown in Fig 9A, GapMind identified a high-confidence path for 85% of carbon sources that support growth, and for just 24% of other carbon sources. For carbon sources that are utilized, transport steps on the best path are more likely to be low- or medium-confidence than enzymatic steps are (5.9% vs. 2.6%, P = $1.5 \cdot 10^{-7}$, Fisher exact test). We suspect that this reflects the greater difficulty of annotating transporters by similarity, and also the greater difficulty of identifying transporters from fitness data because they are often genetically redundant (see below).

Cases where the organism doesn't grow, despite having high-confidence candidates for all of the necessary steps, could indicate inadequate expression of those genes. For example,

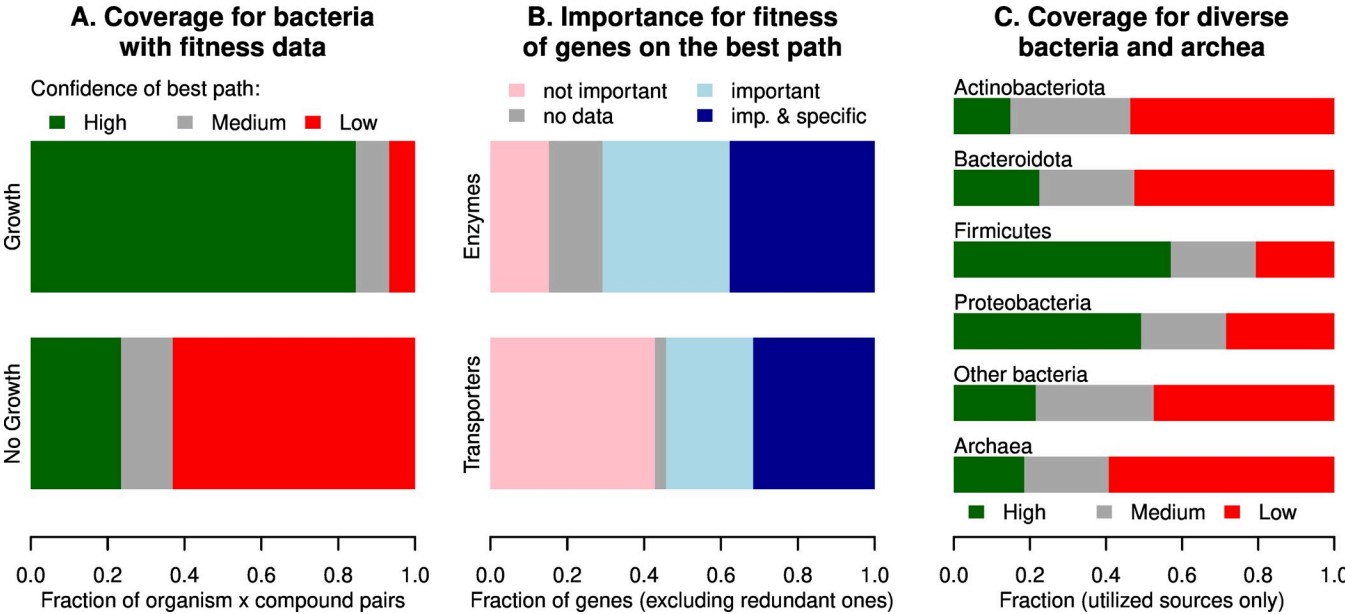

**Fig 9. Quality of GapMind's results.** (A) Confidence of the best path for utilized and non-utilized carbon sources, across 57 carbon sources and 29 heterotrophic bacteria with fitness data. A path is low confidence if it has any low-confidence steps (and similarly for medium confidence). Proportions are from 700 utilized cases and 953 non-utilized cases. (B) Whether high-confidence and non-redundant genes on the best path were important for fitness. Proportions are from 962 genes that encode transporters and 1,254 genes that encode enzymes. Genes lack fitness data if they have insufficient coverage by transposon insertions (usually these are essential or short genes). A phenotype is "specific" if the gene has little phenotype in most other conditions [4]. (C) Confidence of the best path for utilized carbon sources across diverse bacteria and archaea. The phylum assignments are from the Genome Taxonomy Database [49], and "other phyla" includes 14 phyla with less than 100 organism x compound pairs each. There were 54 pairs for archaea.

EcoCyc reports that *E. coli* K-12 does not grow aerobically at 37˚C on 11 of the carbon sources in GapMind, despite containing all of the proteins necessary for their uptake and catabolism. These compounds are arginine, asparagine, aspartate, cellobiose, citrate, ethanol, glutamate, lysine, proline, putrescine, and L-serine. Of the eight nitrogen-containing compounds, seven (all except glutamate) support the growth of *E. coli* as the sole source of nitrogen, which confirms that they are taken up and metabolized.

We also used the fitness data to check if GapMind selected the correct genes for consuming each carbon source. We considered steps that were on the best path, and which had just one high-confidence candidate, because otherwise the genes for the step might be genetically redundant. We analyzed genes that encode enzymes and transporters separately.

When fitness data is available for that gene and condition, 82% of genes that encode enzymes were important for fitness in the condition (Fig 9B). To understand why some of these genes were not important for fitness, we examined a random sample of 20 cases. In 12 of the 20 cases, GapMind identified another high-confidence path as well. For example, in *Shewanella loihica* PV-4, acetate might be converted to the central metabolite acetyl-CoA by acetyl-CoA synthase *(acs)* or else by acetate kinase (in reverse) and phosphate acetyltransferase (*ackA* and *pta*). *E. coli* K-12 uses both pathways to consume acetate [50], so the lack of a phenotype for *ackA* in *S. loihica* could indicate genetic redundancy with *acs*. More broadly, if GapMind identifies two high-confidence pathways, it arbitrarily chooses the one with more steps. (Our intuition is that one step might be annotated erroneously, but the presence of several steps is unlikely unless the pathway is present.) GapMind might guess wrong, or the two pathways might be genetically redundant. For another 6 of the 20 cases we examined, genes for other

steps on the selected path were important for fitness during growth on the carbon source, which suggests that GapMind selected the correct path.

For genes that encode transporters on the best path, and for which fitness data is available, 56% were important for fitness in the condition (Fig 9B). We examined a random sample of 20 cases where the transporter gene was not important for fitness. In most of those cases (18/20), GapMind identified another high-confidence transporter as well, so the genes for the two types of transporters might be genetically redundant. Overall, enzymes and transporters that are part of GapMind's best path for consuming a compound are usually important for fitness during growth with that compound as the sole source of carbon and energy, and most of the exceptions could be due to genetic redundancy.

## Increased coverage of catabolism in diverse bacteria and archaea

Fitness data for these 29 heterotrophic bacteria was used to improve GapMind, so our analyses so far show the best-case performance. As a more realistic test, we examined GapMind's results for diverse bacteria and archaea from the IJSEM database [14]. Overall, GapMind found high-confidence paths for 38% of utilized compounds, and it found medium-confidence paths for another 25% of utilized compounds (Fig 9C). Among the α,β,γ-Proteobacteria and Firmicutes, which are relatively well studied, GapMind found high-confidence paths for 51% of utilized compounds, while in other microbes, GapMind found high-confidence paths for just 20% of utilized compounds, which is significantly less (P = 5 · 10$^{-43}$, Fisher exact test). Even for the α, β,γ-Proteobacteria, which account for 26 of the 29 bacteria with fitness data that were used to improve GapMind, the coverage of utilized carbon sources by high-confidence paths was much lower for bacteria from IJSEM than for the bacteria with fitness data (51% vs. 87%). Much remains to be discovered about the catabolism of these carbon sources.

A "naive" version of GapMind that uses only proteins from curated databases, and does not take advantage of the fitness data or our curation of the literature, finds high-confidence paths for just 27% of utilized compounds (instead of 38%), and finds medium- or high-confidence paths for just 53% of utilized compounds (instead of 63%). In other words, the additional biological knowledge in GapMind helps to explain about 10% of carbon catabolism in diverse bacteria and archaea.

## Conclusions

We suspect that for diverse bacteria, we simply do not know enough to make predictions about what carbon sources they can use. For the 62 compounds whose catabolism is represented in GapMind, and across diverse bacteria and archaea that utilize the compound, GapMind finds a complete path, with at least a medium confidence candidate for each step, for 63% of cases.

Rather than trying to predict a microbe's growth capabilities from its genome sequence, GapMind annotates potential pathways. These annotations help us examine the microbe's potential capabilities and can highlight gaps in our knowledge. Indeed, by using genetic data to explore the gaps in 29 heterotrophic bacteria, GapMind helped us identify hundreds of diverged transporters and enzymes. We also identified a novel pathway for glucosamine utilization and putative novel families of citrullinases, 2-deoxy-5-keto-D-gluconate 6-phosphate aldolases, 3'-ketolactose hydrolases, and oxepin-CoA hydrolases.

The biology we discovered while working on GapMind led to significant improvements in GapMind's results for diverse microbes. The coverage of catabolism by medium-confidence paths improved from 53% to 63%. For example, DUF2090 seems to be the most common form of 2-deoxy-5-keto-D-gluconate 6-phosphate aldolase, and the putative family of citrullinases

may be a common mechanism for the aerobic utilization of citrulline. GapMind for carbon sources captures this knowledge in an easy-to-use tool.

We do hope that accurate predictions of growth capabilities will become feasible. We plan to collect fitness data from more diverse bacteria, which should help to fill many of the gaps, and for more carbon sources. It will also be important to have a large dataset that includes cases where the compound does not support growth, as well as utilized compounds. This would allow us to identify steps that are less important for prediction. For example, transporters or sugar kinases are often challenging to annotate, while some catabolic enzymes are easier to annotate. Given a large dataset with negative cases, it should be straightforward to estimate a weighting for each step.

Another possibility is that for traits that are highly conserved, phenotypes could be predicted from observations for related organisms, instead of focusing on what genes the genome contains. We are not sure if this will be useful for carbon source utilization, because bacteria with similar 16S ribosomal RNA sequences often have quite different carbon source utilization capabilities [2].

## Materials and methods

### Data sources

We obtained the characterized subset of Swiss-Prot [10], BRENDA [11], MetaCyc [7], CAZy [12], CharProtDB [51], and EcoCyc [50] via the PaperBLAST database [9], as described previously [6]. The PaperBLAST database was downloaded in May 2020. For this study, we also incorporated the experimentally-characterized subset of the transporter classification database (TCDB) [13] into PaperBLAST and GapMind. The TCDB fasta file was downloaded in March 2020 and the TCDB web site was queried programmatically in April 2020. Proteins from TCDB were considered to be characterized if they were annotated with a substrate, were linked to a reference, had a description, and the protein was not described as putative or uncharacterized. If any protein from a multi-component transport system was considered characterized, then all of the proteins in the system were retained.

Carbon source utilization data for the 29 heterotrophic bacteria with fitness data was taken from our previous studies [4,8]. For *Bacteroides thetaiotaomicron* VPI-5482, we checked the original growth curve data to verify that the compounds that did not have fitness assays did not support growth as the sole source of carbon. However, we discovered that our original stock solutions for sucrose and D-mannitol were problematic. In particular, *E. coli* BW25113 is a K-12 strain (closely related to MG1655) and should not be able to grow on sucrose. In M9 media made with our original stock solution of sucrose, *E. coli* BW25113 grew, but in media made with a fresh stock solution, it did not. Similarly, growth of *E. coli* on mannitol should require the phosphotransferase uptake protein MtlA and the mannitol 1-phosphate dehydrogenase MtlD ([52]; data of [53]). In our original fitness assays for *E. coli*, *mtlA* and *mtlD* were not important for growth on mannitol; instead, *manX* and *manY*, which encode the mannose phosphotransferase system, were important. When we repeated these experiments with a fresh stock solution for D-mannitol, we found that *mtlA* and *mtlD* were important for fitness, and *manX* and *manY* were not. Because of these problems, we did not include our prior data for mannitol or sucrose.

Fitness data for 29 heterotrophic bacteria was taken from [4,8,45], except that prior data for mannitol and sucrose were ignored. We also analyzed data for *Pseudomonas putida* KT2440 ([54]; Mitchell Thompson and Matthias Schmidt, personal communication) and for *Burkholderia sp.* OAS925 (Marta Torres, personal communication). Fitness data was viewed in the Fitness Browser (http://fit.genomics.lbl.gov/)

Carbon source utilization data for diverse bacteria and archaea was obtained from the IJSEM database (version 1.0, downloaded in April 2019; [14]). We linked these records to genome sequences from RefSeq by matching the genus and strain identifiers. Only genomes with at most 50 scaffolds were considered. This left us with 1,819 pairs of organisms and utilized carbon sources, which cover 45 of GapMind's 62 compounds, 224 bacteria, and 13 archaea. We obtained the predicted protein-coding genes from RefSeq.

## Pooled mutant fitness assays

We collected new fitness data for the utilization of D-mannitol, sucrose, L-citrulline, L-ornithine, or phenylacetate by *Cupriavidus basilensis* FW507-4G11, *Dinoroseobacter shibae* DFL-12, *Escherichia coli* BW25113, *Herbaspirillum seropedicae* SmR1, *Paraburkholderia bryophila* 376MFSha3.1, *Phaeobacter inhibens* BS107, *Pseudomonas fluorescens* FW300-N1B4, *P. fluorescens* FW300-N2C3, *P. fluorescens* FW300-N2E2, *P. fluorescens* FW300-N2E3, *P. simiae* WCS417, or *Shewanella* sp. ANA-3. These pools of randomly-barcoded transposon mutants were described previously [4,45,55], and fitness assays were performed as described previously [55]. Briefly, each pool of transposon mutants was recovered from the freezer in rich media with kanamycin until it reached mid-log phase. Some of this initial "Time0" sample was saved. The mutant library was then inoculated at $OD_{600} = 0.02$ into a defined medium with the compound of interest as the sole source of carbon, at a concentration of between 5 and 20 mM. The defined media also included ammonia as a nitrogen source, other mineral salts, and vitamins. The culture was grown aerobically at 30°C until saturation in a Multitron shaker. Genomic DNA was extracted and barcodes were amplified with one of 96 different primer pairs; both sides of these primers contain unique sequences to ensure accurate demultiplexing. (The sequences of the P1 primers are available from primers/barseq3.index2 in the source code; the sequences of the P2 primers are unchanged from [55].) PCR products were combined (up to 96 samples) and sequenced using Illumina HiSeq 4000.

The fitness data was analyzed as described previously [55]. Briefly, the fitness of a strain is the log2 ratio of the count in the experimental sample (after growth in the media of interest) versus the count in the Time0 sample, normalized so that the median strain fitness is zero. The fitness of a gene is the weighted average of the fitness of strains with insertions in the central 10–90% of that gene. Gene fitness values are also normalized to correct for the effect of chromosomal position (because copy number near the origin of replication is higher in faster-growing cells). Finally, gene values are normalized so that the mode of the distribution is at zero. The source code for these analyses is available at https://bitbucket.org/berkeleylab/feba; we used statistics version 1.3.1. The fitness data is available in the Fitness Browser (http://fit.genomics.lbl.gov) and is archived at https://doi.org/10.6084/m9.figshare.16913530.v1.

## Curating pathways

To identify known pathways for the catabolism of each compound, we relied primarily on MetaCyc. We became aware of a few additional pathways by running PaperBLAST on genes that were important for utilizing the compound, or by using Google scholar. In the GapMind website, each pathway is linked to the MetaCyc page or to a publication.

In general, pathways were only included if all of the metabolic transformations are known and are linked to protein sequences, and are reported to occur in bacteria or archaea. However, a few pathways with one missing reaction were included: deoxyribonate oxidation involves an unknown glyceryl-CoA hydrolase; aerobic oxidation of benzoyl-CoA involves an unknown 3,4-dehydroadipyl-CoA isomerase (benzoyl-CoA is an intermediate in phenylacetate degradation); anaerobic degradation of benzoyl-COA involves an unknown 3-hydroxypimeloyl-CoA

dehydrogenase; and glutamate utilization via (S)-citramalate involves an unknown (S)-citramalate CoA-transferase. Pathways that we omitted due to a lack of knowledge include: degradation of cellobiose, maltose, or sucrose by the 3-ketoglycoside pathway; degradation of arginine via 5-amino-2-oxopentanoate; degradation of deoxyinosine via a nucleosidase; degradation of fructose via phosphofructomutase; fermentation of glutamate via 5-aminovalerate; degradation of tryptophan via anthraniloyl-CoA monooxygenase/reductase; degradation of tryptophan via indole and anthranilate; and degradation of tyrosine via 4-hydroxyphenylpyruvate oxidase.

## Curating transporters and enzymes

To identify characterized transporters for each compound, we automatically combined candidates from MetaCyc's transport reactions, TCDB's substrate descriptions, and characterized proteins from other databases whose descriptions include the compound as well as the terms transport, porter, import, permease, or PTS system. Many compounds were described by multiple terms and also by MetaCyc compound identifiers: for instance, to find transporters for L-fucose, we used the terms L-fucose, L-fucopyranose, CPD-10329, CPD0-1107, and CPD-15619. The results were checked manually.

To identify proteins for each enzymatic reaction, we primarily used enzyme classification (EC) numbers, which are linked to protein sequences by the curated databases. If any TIGR-FAMs are annotated with that EC number [15], then GapMind also uses TIGRFAM's models and HMMer to find candidates for that step.

GapMind does not consider which compartment the reaction occurs in: for example, cellobiose utilization might involve a periplasmic cellobiase and then uptake of glucose, or uptake of cellobiose and then a cytoplasmic cellobiase. Since GapMind does not attempt to predict the subcellular localization of the candidate proteins, any cellobiase it identifies is (unrealistically) assumed to participate in either pathway. On the GapMind website, the page for a candidate does include a link to analyze the protein's sequence with PSORTb 3.0, which predicts protein localization [34].

## Improvements to the GapMind software

To help us define each step, we built a "curated clusters" tool (available at https://papers.genomics.lbl.gov/cgi-bin/curatedClusters.cgi?set=carbon). This tool clusters the curated protein sequences that match a search term or are included in a step definition. It can also cluster the potential transporters for a compound into families of similar transporters. By default, it clusters at 30% identity and 75% alignment coverage (both ways), but this can be changed. In particular, many ABC transporters contain two permease subunits that are similar to each other; to separate the two subunits, we usually clustered these at 40% identity.

The clustering tool is particularly useful for annotating multi-protein transporters and enzymes. To highlight transporters or enzymes that are likely to be heteromeric, the curated clusters tool relies on the explicit complexes in MetaCyc and TCDB; the SUBUNIT field of Swiss-Prot entries; or terms such as "subunit" or "component" in the description.

The clustering tool also helps to identify annotation errors in the source databases. We checked any sequences that do not cluster with other sequences for that query, are annotated by only one database, and have unexpected domain content. (The clustering tool shows the domain content for each protein, using hits from PFam [20].) We identified errors in BRENDA and MetaCyc and notified the curators.

Another feature of the clustering tool is to find other curated sequences that are similar to one of the proteins that is associated with the step, but was not included in the step definition. This sometimes identifies proteins that have the same function but were initially missed due to

inconsistent annotation. Also, if enzymes from this family are known to be somewhat nonspecific, then we "ignored" similar enzymes that are reported to act on a slightly different substrate (but may also act on the substrate of interest). Usually, GapMind will consider a candidate to be lower confidence if it is overly similar to a protein with another function, but any similarity to "ignored" sequences associated with that step is not penalized.

As another way to simplify curation, GapMind now allows steps or rules from one pathway to be imported into another pathway. This ensures that any improvements to a step definition can be recorded in just one place.

GapMind for carbon sources considers a much larger number of candidate steps than GapMind for amino acid biosynthesis. To ensure a reasonable running time, we reduced the number of candidates considered for each step. For each step x genome pair, GapMind now considers only the four top candidates by bit score, and it considers at most two candidates with alignments of under 40% identity. This reduces the running time because GapMind uses ublast to compare every candidate it finds against its database of characterized proteins. To speed the analysis of these results, GapMind now considers only the top eight characterized hits (by bit score) for each candidate. Also, GapMind now uses sqlite3 databases instead of tab-delimited files to access the database of curated proteins, the proteins associated with each step, and other information about the steps and rules.

## Thresholds for high-confidence candidates

For candidates from ublast, the most important criteria for classifying them as high-confidence are at least 40% identity to a characterized homolog with at least 80% alignment coverage. To see how these thresholds perform for catabolic enzymes and transporters, we tested them on the characterized proteins that are associated with steps in GapMind. Specifically, we compared each of these proteins to all other characterized proteins (using ublast), and asked if the best hit met the thresholds for a high-confidence candidate. If it did, we asked if the best hit was also associated with that step. Of the 6,742 characterized enzymes in GapMind for carbon sources, 5,105 had a hit above 40% identity and 80% coverage. For those sequences, the best hit was associated with the same step in 85% of cases. Of the 1,393 characterized transporters in GapMind, 1,295 had a hit above these thresholds, and the best hit was associated with the same step in 61% of cases. The high-confidence candidates from diverse bacteria and archaea tend to have lower %identity to their best hits than these characterized proteins do (median 57% instead of 72%), so accuracy would be lower in practice. For best hits of 40–50% identity (near the threshold), the accuracy was 73% for enzymes and 32% for transporters. If we considered the best hits of characterized proteins from bacteria and archaea only, we got similar results (71% and 29%, respectively for best hits of 40–50% identity).

The expectation (E) value of the alignment is not considered by GapMind. At a given level of sequence divergence, $\log(E)$ scales linearly with alignment length, so we believe that %identity is a more informative metric. Also, GapMind runs ublast with a relatively lax threshold ($E \leq 0.01$ against the database of proteins that are linked to steps), but because of the constraints on alignment identity and coverage, this has little effect on the results. In the analysis of 237 diverse bacteria and archaea from IJSEM, the weakest alignments for high-confidence candidates on the best path had scores of 46.6 bits, which corresponds to $E = 3.7 \cdot 10^{-5}$ if comparing to the database of all characterized proteins.

## Comparison to fitness data

If GapMind did not identify a high-confidence path for a compound, and we had fitness data for the compound, then we attempted to find candidates using the fitness data. We found

most of these candidates by using specific phenotypes: genes that are important for fitness in that condition but not in most other experiments [4]. To find genes with weaker or broader phenotypes, we sometimes used cofitness with genes in the pathway or scatter plots of gene fitness during growth in this condition versus growth in another carbon source. Potential functions of the candidate genes were checked with PaperBLAST, which finds papers about similar proteins [9]. If we found a plausible candidate for a step and mutants of that gene had the correct phenotype, we added the protein to an existing step definition or added a new step.

## Software versions

GapMind uses ublast from usearch v10.0.240_i86linux32 to find protein similarities and uses HMMer 3.3.1 for HMM searches. We also used NCBI BLAST 2.2.18. For statistical analyses and plotting, we used R 3.6.0.

## Supporting information

**S1 Fig. Utilization of arginine, citrulline, ornithine and proline by *Pseudomonas fluorescens* FW300-N2E3.**
(PDF)

**S2 Fig. Utilization of arginine, citrulline and ornthine by *Pseudomonas simiae* WCS417.**
(PDF)

**S3 Fig. Utilization of citrulline, ornithine, and proline by Phaeobacter inhibens DSM 17395 (BS107).**
(PDF)

## Acknowledgments

We thank Robin Herbert and Trenton Owens for providing fitness data from *Paraburkholderia bryophila* 376MFSha3.1. We thank Mitchell Thompson, Matthias Schmidt, and Marta Torres for providing pre-publication access to fitness data from *Pseudomonas putida* and *Burkholderia sp.* OAS925.

## Author Contributions

**Conceptualization:** Morgan N. Price, Adam M. Deutschbauer, Adam P. Arkin.

**Data curation:** Morgan N. Price.

**Funding acquisition:** Adam M. Deutschbauer, Adam P. Arkin.

**Investigation:** Morgan N. Price, Adam M. Deutschbauer.

**Methodology:** Morgan N. Price.

**Resources:** Adam M. Deutschbauer, Adam P. Arkin.

**Software:** Morgan N. Price.

**Supervision:** Adam P. Arkin.

**Visualization:** Morgan N. Price.

**Writing – original draft:** Morgan N. Price.

**Writing – review & editing:** Morgan N. Price, Adam M. Deutschbauer, Adam P. Arkin.

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
