## [Decision Letter · Decision Letter 0]

27 Jan 2022

Dear Dr Price,

Thank you very much for submitting your Research Article entitled 'GapMind for Carbon Sources: Automated annotations of catabolic pathways' to PLOS Genetics.

The manuscript was fully evaluated at the editorial level and by independent peer reviewers. The reviewers appreciated the attention to an important problem, but raised some substantial concerns about the current manuscript. Based on the reviews, we will not be able to accept this version of the manuscript, but we would be willing to review a much-revised version. We cannot, of course, promise publication at that time.

If you decide to revise the manuscript for further consideration at PLOS Genetics, please aim to resubmit within the next 60 days, unless it will take extra time to address the concerns of the reviewers, in which case we would appreciate an expected resubmission date by email to plosgenetics@plos.org.

[LINK]

We are sorry that we cannot be more positive about your manuscript at this stage. Please do not hesitate to contact us if you have any concerns or questions.

Yours sincerely,

Bernhard O. Palsson

Guest Editor

PLOS Genetics

Lotte Søgaard-Andersen

Section Editor: Prokaryotic Genetics

PLOS Genetics

Reviewer's Responses to Questions

**Comments to the Authors:**

Reviewer #1: Summary:

Given the genome of a bacterial or archaeal organism, the web-based GapMind tool uses the PaperBLAST tool (which searches open-access scientific articles) to compare the genome with known sequences for catabolic and transport enzymes. The tool focuses on identifying transport and catabolic enzymes for an array of 62 carbon sources. Results are annotated with confidence categories (high, medium, low) based on the PaperBlast results for identity and coverage. Predicted catabolic pathways and essential genes for catabolic pathways were tested in vivo for several of the 29 bacterial species results studied here. This tool will be a valuable resource (as a starting point) for reconstruction of metabolic models for poorly studied species. This tool could also be of use to in vivo researchers who wish to begin learning about or investigating the metabolism of a poorly studied species. Despite its potential usefulness, there are a few major concerns about the tool in its current form which should be addressed, particularly the lack of verification or analysis for archaeal species, which need be addressed before this work can be considered for publication.

General reviewer comments:

Major Concerns:

1) Match thresholds. The thresholds for high, low, and medium confidence appear arbitrary, yet are important to the results of the GapMind tool. Please provide a justification so to why (for instance) a 40% identity match with 80% confidence is sufficient a high-confidence label? For instance, for some benchmark enzyme like alcohol dehydrogenase, what fraction of your experimentally characterized alcohol dehydrogenase proteins would return a high-confidence result when compared to each other. Alternatively, are your thresholds some type of established standard? If so, please cite a source which uses this standard. In this or related works, have multiple threshold values been compared to verify the impact it would have in their results? If so, it would be interesting to see those results described and discussed.

2) Spurious alignments. The chosen E-value cut-off is 0.001, which is probably too high. Please justify why this value was selected. Additionally, for the alignments made to represent transcriptional/translational units, open reading frames need to first be determined, but this is not addressed in the text. Can the authors please elaborate on how they determined open reading frames?

3) Fitness to analyze archaea genomes and catabolic pathways. GapMind is designed to be used for bacteria and archea; however, these are two different domains of life and can be very different in metabolism, enzymes, and enzyme sequences. Yet, only bacteria were extensively examined in this paper (save for a brief analysis reported in Figure 9C). In this analysis, it was found that a relatively low fraction of carbon sources used by archaea (about 20%) were marked as high confidence pathways. What is the justification for this tools fitness when applied to archaea? Some potential justification could be given in Table 1 if there was a breakdown of the domain sources of the proteins used by GapMind (e.g., what fraction of these proteins are archaeal and bacterial), though this would not be sufficient in and of itself to address this concern. This lack of training is not addressed in the conclusion to this manuscript when plans to incorporate more bacterial fitness data is discussed.

4) Approach to identifying catabolic enzymes. The approach to identifying catabolic enzymes taken by the GapMind tool is to identify catabolic pathways via blast analysis. However, even high-identity blast results do not necessarily mean that the sequence maps to that enzyme function (consider the results reported in figures 8A and 8B of this manuscript, particularly the high error rate at under 60% identity). Perhaps more important than whole enzyme identity is the identity of critical residues for enzyme structure and active site. This is something that might be brought to light using Hidden Markov Models (HMMs). While “candidates are identified by using ublast or HMMer” (line 125), the role of these techniques in identifying a list of catabolic or transport candidate enzymes is unexplained. As written, the manuscript then seems to indicate that residue identity is the most important factor in whether or not a protein has the function of a particular enzyme. The question of this concern is why is identity chosen as more important than conservation of important residues or HMMs?

5) Use of fitness data. In this work, fitness data was used both to improve – line 54 –tool and to support its predictions – line 69 - of the GapMind i.e., both for training and validation. Then how does the tool perform using data on which it is not trained? Would there be a prerequisite to train the tool to the data it is to analyze? These results seemingly describe a confirmation of the results this tool was designed to show. Can the authors address this issue in their discussion?

Minor Concerns:

1) Web interface. As a test of the tool, the reviewer used the web interface to investigate the small carbon molecule catabolism in E. coli. The web interface was, overall, easy to use (up to the point of the report), but a few improvements and clarifications need to be made:

a. In the “About GapMind” section, it is stated that the tool uses “ublast” but when one clicks on the various links of a high- or medium- confidence result, they are given PaperBLAST reports. The web tool does not make it clear as to what “ublast” is and what it is used for. Is this a separate tool from PaperBLAST, or merely some interface which PaperBlast is performed through?

b. For some low-confidence results, there are no reported BLAST results. However, by the definition of “low confidence” which is given in the interface, there must be a blast hit with at least 50% coverage to be reported as “low confidence”. Where might this blast hit be found or recovered? Further, while high- and medium- confidence matches will provide a link to pfam (as an added layer to evaluate the strength of the match), this would be most useful for low-confidence matches such as shown in the screenshot, yet because the blast result is not reported, the pfam analysis could not also be performed.

c. The caveats described in the results and discussion section “overview of GapMind for carbon sources” should be mentioned somewhere on the report page (for instance after the list of the carbon catabolic steps) so it is clear to users what GapMind does not do.

d. For the blast reports, many would be interested not only in identity, but e-value (as this is a common metric used for the quality of a match) and percent positives (that is, amino acids substituted with similar-type amino acids).

e. Once the report is generated, having website links like “home”, “return to results”, and similar webpage navigation tools would make the webpage easier to navigate once the results have been produced.

f. When looking at the PaperBLAST results, it is initially difficult to find the query sequence used (because it is located near, but not at, the bottom of the results page).

2) Transporter ambiguity. How does GapMind treat transporters which are annotated to transport several different substrates or are ambiguous about substrates transported (for instance, the gene Cthe_1862 for Clostridium thermocellum which is annotated as “multiple sugar transport system ATP-binding protein”). Is it permissive (e.g., assuming such a system could transport anything marked as a sugar) or strict (e.g., ignoring this gene as it does not specify exact substrates)?

3) Association between proteins. Please elaborate on why all the proteins discussed in the text are assumed to be distantly-related (e.g. divergent evolution from a single origin) as opposed to different events, such as convergent evolution. Is this true for each protein instance that is described?

4) L57: Please clarify what is meant by “initial version” of GapMind? Does that refer to an iterative process or to a prototype of the algorithm created?

5) L69: Please clarify what is meant by “… GapMind usually selects the correct pathway…”? Was the success rate of GapMind estimated? If so, why isn’t it referred to explicitly?

6) L300-302: Bioinformatics evidence supports the authors’ proposition that DUF2090 is a 2-deoxy-5-keto-D-gluconate 6-phosphate aldolase but does not confirm it. Experimental evidence would be necessary to confirm such a proposition.

7) L541: Please justify why GapMind chooses the pathway with more steps, when that could possibly represent a more burdensome pathway for the cell to execute the same function?

Concerns of grammar, clarity, spelling, and similar:

1) The manuscript appears to be well edited, and the writing is clear.

Reviewer #2: The paper describes the use of genome sequence data, extensive mutant growth screening data, and their GapMind tool to characterize interesting catabolic functions in several specific bacteria.

Major comments:

- Novelty & framing of the paper’s key contributions. The paper is set up (given title and abstract) as the description of a new tool (GapMind) for predicting catabolic pathways given a genome. However, it’s unclear how this database/tool is different than the previously published GapMind resource cited in the manuscript (Price et al. mSystems 2020) other than the focus on carbon catabolism vs. amino acid synthesis. In addition, the manuscript discriminates between the annotation of catabolic pathways and the network model-derived predictions of the catabolic functions of an organism, arguing that predicting a catabolic function is difficult because of variance in protein expression or perhaps because of erroneous annotations. However, again, working through the differences between annotating catabolic pathways and model-driven predictions of catabolic functions is not really the focus of the paper.

The real contribution of the paper seems to be how the use of the GapMind database can help provide context to the large mutant growth screening data to enable stronger predictions for functional annotation of catabolic pathways. This objective is a nice focus, the authors make a nice contribution on this point, and the new biology discussed as an outcome of these analyses is a nice contribution. All this to say that the manuscript should be revised to better frame these more valuable contributions of the paper.

- GapMind identifies a path with “all high-confidence steps” or that has “no low-confidence steps”, or that has the “highest total score”. The manuscript would benefit from an analysis of the differences between these thresholds and sensitivity to all these types of thresholds that are used in assigning confidences and in “selecting” a pathway.

- The reference pathways included in the GapMind database include those for which a complete pathway is known or for which one step is missing. The robustness of the analysis in the paper to this decision should be considered. What happens if pathways are included with two missing steps? Zero missing steps?

- For the last section of the Results on assessing the quality of GapMind’s results, it is unclear what is actually being assessed. It seems the validation is grow/no grow data of 29 bacteria on 57 carbon sources, but a key component of GapMind predictions are which pathways are used for a given catabolic function and the grow/no grow data the authors reference do not seem to have that kind of resolution.

Minor comments:

- While there are many new annotation assignments presented in this manuscript, it almost becomes a little unwieldy and the manuscript would benefit from a more concise summary and more focused description of fewer specific stories.

- The black text on the dark blue background in several figures is difficult to see.

Reviewer #3: Title: GapMind for Carbon Sources: Automated annotations of catabolic pathways

Authors: MN Price, AM Deutschbauer, AP Arkin

Summary

The authors developed a computational tool GapMind, and web app, that automatically annotates catabolic pathways for bacterial and archaeal genomes.

The authors previously developed GapMind to annotate amino acid biosynthesis pathways. The present manuscript is an upgrade to GapMind for carbon source catabolic pathways. Improvements are made to GapMind’s algorithms and databases, including the authors’ newly generated high-throughput experimental (transposon mutant fitness) data as resources. Furthermore, “curated clusters” was built to define each step.

GapMind relies on various databases of pathways, transporters, and enzymes. It also relies on high-throughput genetic data and literature knowledge for improved accuracy. It also relies on “curated clusters” tool that clusters the protein sequences that are part of a step in the pathway. Protein similarity is computed using ublast and hmmer. Classification of predicted proteins as low-, medium-, and high- confidence proteins helps users focus on proteins that need most attention. The tool is fast at prediction (30 seconds per genome).

GapMind is a genome sequence-based annotation tool. Because of this limitation, only presence or absence of pathways is detected, but condition-specific utilization of pathway is not predicted by the tool especially when redundant pathways are identified in an organism. It also does not consider the subcellular localization of the candidate protein.

Overall, the study is fascinating and will aid the research of uncharacterized organisms and the identification of novel pathways of substrate catabolism. I believe the research is of interest to the community, and with some revisions or clarifications, the manuscript can be publishable. Please see my comments below for further details.

Major comments

1. Inclusion of controls

a. For the role of glucosamine utilization (Page 10 to Page 12), the investigators have not used any control. Even though the analysis is good, inclusion of a substrate control such as glucose would strongly support the conclusions.

2. Figures can be improved.

a. There are fitness assay results that use one or two experiments. In such cases, taking averages might not make sense. But there are many instances where the authors decided to report individual numbers. Despite being more informative, the figures do not look engaging. I suggest revision of the visualization especially for fitness data.

b. Resolution of some of the figures is poor. If possible, they should be improved.

Minor comments

1. Improvements in the figures

a. Figure 1 (Page 6): Visually it can be better. I would also modify the color scheme (especially changing green or red).

b. Figure 6 (Page 20): The grey rows are not properly explained in the figure caption. If the data is absent for the gene, these rows can be entirely removed.

c. Figure 4 (Page 14):

i. The figure caption can be improved (especially for figure E).

ii. CtlX should be ctlX as it is a gene, and not a protein in this context.

d. Supplementary figures: Figure captions can be improved.

2. Quantitative tests for determination of differences

a. For the experiments involving ketolactose hydrolase, statistical tests for comparison between different substrates (for gene fitness) would be better.

3. Line 386 to 389 (page 23): This information needs further elaboration as I did not understand why authors chose to bring up 3-ketohexose/3-ketoglucose utilization.

4. Line 393 to 397 (page 23): The sentence should be rephrased as it appears to be grammatically inconsistent.

5. Line 540 to 542 (page 31): The authors suggest that if two high confidence pathways are identified, the tool arbitrarily chooses the one with more steps. This is counterintuitive to metabolic pathway algorithms that use parsimonious approach. If a minimum number of steps approach is taken instead, will accuracy be affected negatively?

**Have all data underlying the figures and results presented in the manuscript been provided?**

Reviewer #1: Yes

Reviewer #2: **No: **There are a couple instances where the referenced data is from "personal communications".

Reviewer #3: Yes

PLOS authors have the option to publish the peer review history of their article (what does this mean?). If published, this will include your full peer review and any attached files.

Reviewer #1: No

Reviewer #2: No

Reviewer #3: No

---

## [Decision Letter · Decision Letter 1]

18 Mar 2022

Dear Dr Price,

We are pleased to inform you that your manuscript entitled "Filling Gaps in Bacterial Catabolic Pathways with Computation and High-throughput Genetics" has been editorially accepted for publication in PLOS Genetics. Congratulations!

Yours sincerely,

Bernhard O. Palsson

Guest Editor

PLOS Genetics

Lotte Søgaard-Andersen

Section Editor: Prokaryotic Genetics

PLOS Genetics

Comments from the reviewers (if applicable):

Please address the minor revisions suggested by reviewers in the final draft uploaded for publication.

Reviewer's Responses to Questions

**Comments to the Authors:**

Reviewer #2: The authors have sufficiently addressed the critical points raised in my initial review.

Reviewer #3: Price et al.’s work on GapMind is interesting, and it will provide researchers with a tool to predict pathways of diverse carbon source catabolization in various organisms. In the manuscript revision, the authors have done a commendable job in addressing my concerns. While I have a couple of minor concerns that should be addressed before publication, I believe the authors have tackled major issues that I found in the original manuscript.

Minor comments:

1. Figure captions remain unchanged:

a. "b. Figure 6 (Page 20): The grey rows are not properly explained in the figure caption. If the data is absent for the gene, these rows can be entirely removed."

We updated the caption to explain that grey means no data (change #14). We thought it best to keep those rows. In panel A, the row is the only indication that glucose 6-phosphate dehydrogenase is in the cluster of glucose utilization genes. And in panel C, we'd have to explain why lacB was not shown, so it seemed simpler to leave it in.

b. "c. Figure 4 (Page 14): i. The figure caption can be improved (especially for figure E). ii. CtlX should be ctlX as it is a gene, and not a protein in this context."

We corrected the caption for 4D and added more detail to the caption for 4E (change #14).

2. (Lines 375-377) In this hypothetical scenario, the expression of both genetically-redundant β-galactosidase genes must depend on lactose oxidation, so we consider it unlikely.

Couldn’t either of the genes depend on lactose oxidation in this hypothetical scenario?

**Have all data underlying the figures and results presented in the manuscript been provided?**

Reviewer #2: Yes

Reviewer #3: Yes

PLOS authors have the option to publish the peer review history of their article (what does this mean?). If published, this will include your full peer review and any attached files.

Reviewer #2: No

Reviewer #3: No

**Data Deposition**

http://datadryad.org/submit?journalID=pgenetics&manu=PGENETICS-D-21-01543R1

**Press Queries**

---

## [Editor Report · Acceptance letter]

30 Mar 2022

PGENETICS-D-21-01543R1 

Filling Gaps in Bacterial Catabolic Pathways with Computation and High-throughput Genetics 

Dear Dr Price, 

We are pleased to inform you that your manuscript entitled "Filling Gaps in Bacterial Catabolic Pathways with Computation and High-throughput Genetics" has been formally accepted for publication in PLOS Genetics! Your manuscript is now with our production department and you will be notified of the publication date in due course.

With kind regards,

Livia Horvath

PLOS Genetics

On behalf of:
